# Therapeutic targeting ERRγ suppresses metastasis via extracellular matrix remodeling in small cell lung cancer

Hong Wang[1,9], Huizi Sun[1,9], Jie Huang[2,9], Zhenhua Zhang[1], Guodi Cai[1], Chaofan Wang[3], Kai Xiao[4], Xiaofeng Xiong [ID][1], Jian Zhang[5], Peiqing Liu [ID][1,6,7], Xiaoyun Lu [ID][3✉], Weineng Feng [ID][8✉] & Junjian Wang [ID][1,6,7✉]

## Abstract

**Small-cell lung cancer (SCLC) is the most aggressive and lethal type of lung cancer, characterized by limited treatment options, early and frequent metastasis. However, the determinants of metastasis in SCLC are poorly defined. Here, we show that estrogen-related receptor gamma (ERRγ) is overexpressed in metastatic SCLC tumors, and is positively associated with SCLC progression. ERRγ functions as an essential activator of extracellular matrix (ECM) remodeling and cell adhesion, two critical steps in metastasis, by directly regulating the expression of major genes involved in these processes. Genetic and pharmacological inhibition of ERRγ markedly reduces collagen production, cell-matrix adhesion, microfilament production, and eventually blocks SCLC cell invasion and tumor metastasis. Notably, ERRγ antagonists significantly suppressed tumor growth and metastasis and restored SCLC vulnerability to chemotherapy in multiple cell-derived and patient-derived xenograft models. Taken together, these findings establish ERRγ as an attractive target for metastatic SCLC and provide a potential pharmacological strategy for treating this lethal disease.**

**Keywords** ERRgamma; Small-Cell Lung Cancer; Extracellular Matrix Remodeling; Metastasis; Therapeutic Target
**Subject Categories** Cancer; Respiratory System

## Introduction

Small cell lung cancer (SCLC) is a high-grade neuroendocrine carcinoma that accounts for approximately 15% of all lung cancers and represents the most aggressive and lethal subtype of lung cancer (Siegel et al, 2022). Compared with non-small cell lung cancer (NSCLC), SCLC is characterized by a higher proliferation rate, early metastasis and poorer prognosis (Rudin et al, 2021). Metastasis is the leading cause of cancer related death, and most SCLC patients are diagnosed with distant metastasis (Rudin et al, 2021). The common sites of SCLC metastasis include the liver, lymph nodes, brain and bones (Rudin et al, 2021). The standard first-line therapy for SCLC consists of chemotherapy (cisplatin or carboplatin with etoposide) in combination with immunotherapy (antibodies to programmed cell death protein 1 (PD1), and antibodies to PD1 ligand 1 (PD-L1)) (Zugazagoitia and Paz-Ares, 2022). However, despite initial sensitivity to therapy, SCLC almost invariably progresses due to therapy resistance, resulting in a median overall survival (OS) of only approximately 12–15 months (Gazdar et al, 2017). Overcoming drug resistance has become the main obstacle and the central issue to be addressed in improving SCLC prognosis. Thus, there is an urgent need to identify novel, tractable therapeutic targets and to develop more effective treatment strategies for SCLC.

The nuclear receptor superfamily, which consists of 48 key transcription factors, is a class of proteins that together represent major therapeutic drug targets for human diseases. These receptors possess conserved ligand-binding domains, making them suitable for ligand-based drug discovery and ideal targets for therapeutic intervention. Estrogen-related receptor gamma (ERRγ, encoded by the ESRRG gene), along with ERRα and ERRβ, constitutes the nuclear receptor subfamily ERRs. ERRs exhibit high sequence similarity to estrogen receptors (ERs), but do not bind to endogenous estrogens or their

[1]School of Pharmaceutical Sciences, Sun Yat-sen University, 510006 Guangzhou, Guangdong, China. [2]Guangdong Lung Cancer Institute, Guangdong Provincial People's Hospital (Guangdong Academy of Medical Sciences), Southern Medical University, 510080 Guangzhou, China. [3]International Cooperative Laboratory of Traditional Chinese Medicine Modernization and Innovative Drug Discovery of Chinese Ministry of Education (MOE), School of Pharmacy, Jinan University, #855 Xingye Avenue, 510632 Guangzhou, China. [4]Precision Medicine Research Center, Frontiers Science Center for Disease-Related Molecular Network, West China Hospital, Sichuan University, 610041 Chengdu, China. [5]Thoracic Surgery Department, The Third Affiliated Hospital of Sun Yat-sen University, No. 600, Tianhe Road, Tianhe District, 510630 Guangzhou, China. [6]National-Local Joint Engineering Laboratory of Druggability and New Drugs Evaluation, Sun Yat-sen University, 510006 Guangzhou, Guangdong, PR China. [7]Guangdong Provincial Key Laboratory of New Drug Design and Evaluation, School of Pharmaceutical Sciences, Sun Yat-sen University, 510006 Guangzhou, Guangdong, P.R. China. [8]Department of Pulmonary Oncology, The First People's Hospital of Foshan, 528000 Foshan, Guangdong, China. [9]These authors contributed equally: Hong Wang, Huizi Sun, Jie Huang. ✉E-mail: luxy2016@jnu.edu.cn; fwneng@fsyyy.com; wangjj87@mail.sysu.edu.cn

derivatives (Heard et al, 2000; Horard and Vanacker, 2003). To date, no endogenous ligand of ERR has been identified, making them members of a subfamily of orphan nuclear receptors (Horard and Vanacker, 2003). Like ERRα and ERRβ, ERRγ exerts its transcriptional activity by binding to the ERR response element (ERRE) as dimer or monomer in a ligand-independent manner; the transcriptional activity of ERRγ is regulated by coactivator or corepressor proteins (Giguère, 2008; Lonard and O'malley, 2007). The role of ERRγ in tumors involves many physiological processes, such as drug resistance, the mesenchymal-to-epithelial transition and cellular metabolism (Chen et al, 2020; Chanda et al, 2023; Choi et al, 2022; Tiraby et al, 2011). ERRγ has been reported to promote migration and metastasis of endometrial cancer cells by targeting S100A4 (Hua et al, 2018). Additionally, in lung adenocarcinoma A549 cells, ERRγ contributes to the bisphenol A (BPA)-induced epithelial mesenchymal transition (EMT) (Ryszawy et al, 2020). However, the function of ERRγ in the pathogenesis of SCLC remains largely unknown.

The extracellular matrix (ECM), an acellular component of the tumor microenvironment (TME), consists primarily of proteoglycans, glycoproteins, matricellular proteins and structural proteins. ECM remodeling, which is characterized by changes in the content, activity and crosslinking of these proteins, is increasingly recognized as a key factor in cancer progression (Cox, 2021). The most common type of oncogenic ECM remodeling involves the enhanced synthesis and deposition of fibrillar collagen accompanied by the expression of remodeling enzymes, leading to an architectural and bioactive environment that supports cancer invasion and metastasis (Winkler et al, 2020). SCLC is surrounded by an extensive stroma of ECM that protects these cancer cells from chemotherapy-induced cell death through the activation of β1 integrins, which leads to the activation of phosphoinositide-3-OH kinase (PI3-kinase) (Hodkinson et al, 2007; Buttery et al, 2004). Integrin β1 can also promote SCLC metastasis by activating downstream focal adhesion kinase/SRC (FAK/SRC) signalling (Zhao et al, 2019). Nevertheless, the molecular mechanism through which the ECM contributes to SCLC progression is still unclear.

In this study, we demonstrated significant overexpression of ERRγ in SCLC tumors, particularly in SCLC tumor cells present in metastases. ERRγ plays a pivotal role as a determinant of extracellular matrix (ECM) remodeling and cell adhesion by directly regulating the transcription of associated genes. The use of an ERRγ antagonist strongly inhibits tumor growth and metastasis in SCLC xenografts via ECM remodeling. Furthermore, we showed that the ERRγ antagonist effectively enhances the sensitivity of resistant tumors to chemotherapy. These findings suggest that ERRγ may be a promising novel therapeutic target for advanced SCLC.

# Results

## ERRγ is overexpressed in SCLC and is associated with SCLC metastasis

To identify the nuclear receptors (NRs) that play crucial roles in SCLC metastasis, we used single-cell RNA sequencing (scRNA-seq) to assess the expression levels of the indicated NRs in normal lung tissues, primary tumors and SCLC metastatic sites (Chan et al, 2021). As shown in Fig. 1A,B, among the NRs, the expression of ESRRG

(the gene encoding ERRγ) was most significantly elevated in metastatic SCLC. To further determine the cell types that contribute to ESRRG expression, we classified the cells into 7 major types: epithelial cells, T cells, macrophages, monocytes, B cells, fibroblasts and endothelial cells (Fig. 1C). As expected, the fraction of epithelial cells was obviously increased in SCLC compared to normal lung (Fig. 1C). Among the cell types, ESRRG was predominantly expressed in epithelial cells, suggesting that ESRRG exerts its regulatory functions largely through tumor cells rather than through immune cells and stromal cells (Fig. 1D,E). In addition, ESRRG expression in epithelial cells was confirmed to be significantly elevated in metastatic SCLC compared with its expression in primary SCLC (Fig. 1F; Appendix Fig. S1A).

To further investigate the potential relationship between ESRRG and SCLC metastasis, a metastasis trajectory from primary tumor cells to metastatic cells was constructed (Appendix Fig. S1B). The expression of ESRRG progressively increased along the trajectory, suggesting a correlation between high ESRRG expression and SCLC metastasis (Fig. 1G). Immunohistochemical (IHC) analysis of primary tumors and metastatic tumors from SCLC patients revealed a strong correlation between the presence of high levels of ERRγ protein and tumor metastasis (Fig. 1H). Furthermore, IHC analysis of another cohort of normal and tumor tissues from patients revealed significant upregulation of ERRγ at the protein level in SCLC tumors compared to normal lung tissues (Appendix Fig. S1C). Moreover, Kaplan–Meier analysis of published tumor datasets revealed that high expression of ESRRG was associated with poor prognosis in patients with SCLC (Fig. 1I) (Liu et al, 2024). These findings suggest that ERRγ, which is primarily expressed in tumor cells, is progressively upregulated during SCLC development, particularly during the progression from primary tumors to metastases.

## ERRγ is a major driver of SCLC cell growth and survival both in vitro and in vivo

To examine the function of ERRγ in SCLC, we first performed shRNA and siRNA knockdown of ESRRG (the gene encoding ERRγ) in multiple SCLC cell lines. As shown in Fig. 2A, knockdown of ESRRG markedly inhibited the growth of SCLC cells, resulting in poor survival of the cells as measured by colony formation and causing pronounced apoptosis, reflected by the activation of caspase3/7 and cleaved PARP1 (Fig. 2B–D; Appendix Fig. S2A–C). To determine whether an elevated level of ERRγ alone is sufficient to promote the growth and survival of SCLC cells, ERRγ was overexpressed in SCLC cells. ERRγ overexpression significantly enhanced cell growth and colony formation (Appendix Fig. S2D–F). We next examined whether pharmacological inhibition of ERRγ activity has strong growth-inhibitory effects on SCLC cells. Similar to the effects of ESRRG knockdown, the ERRγ antagonists DN200434 and GSK5182 strongly inhibited the growth of a panel of SCLC cells (Fig. 2E,F; Appendix Fig. S2G–H). Consistent with the apoptosis induced by ESRRG gene silencing, DN200434 promoted SCLC cell apoptosis (Fig. 2G; Appendix Fig. S2I,J). Considering that 3D organoids may closely mimic the clinical response of tumors to therapeutics, we treated organoids derived from SCLC patient-derived xenografts (PDX) tumors with DN200434. Treatment with DN200434 potently suppressed the growth of the organoids in a dose-dependent manner (Fig. 2H,I).

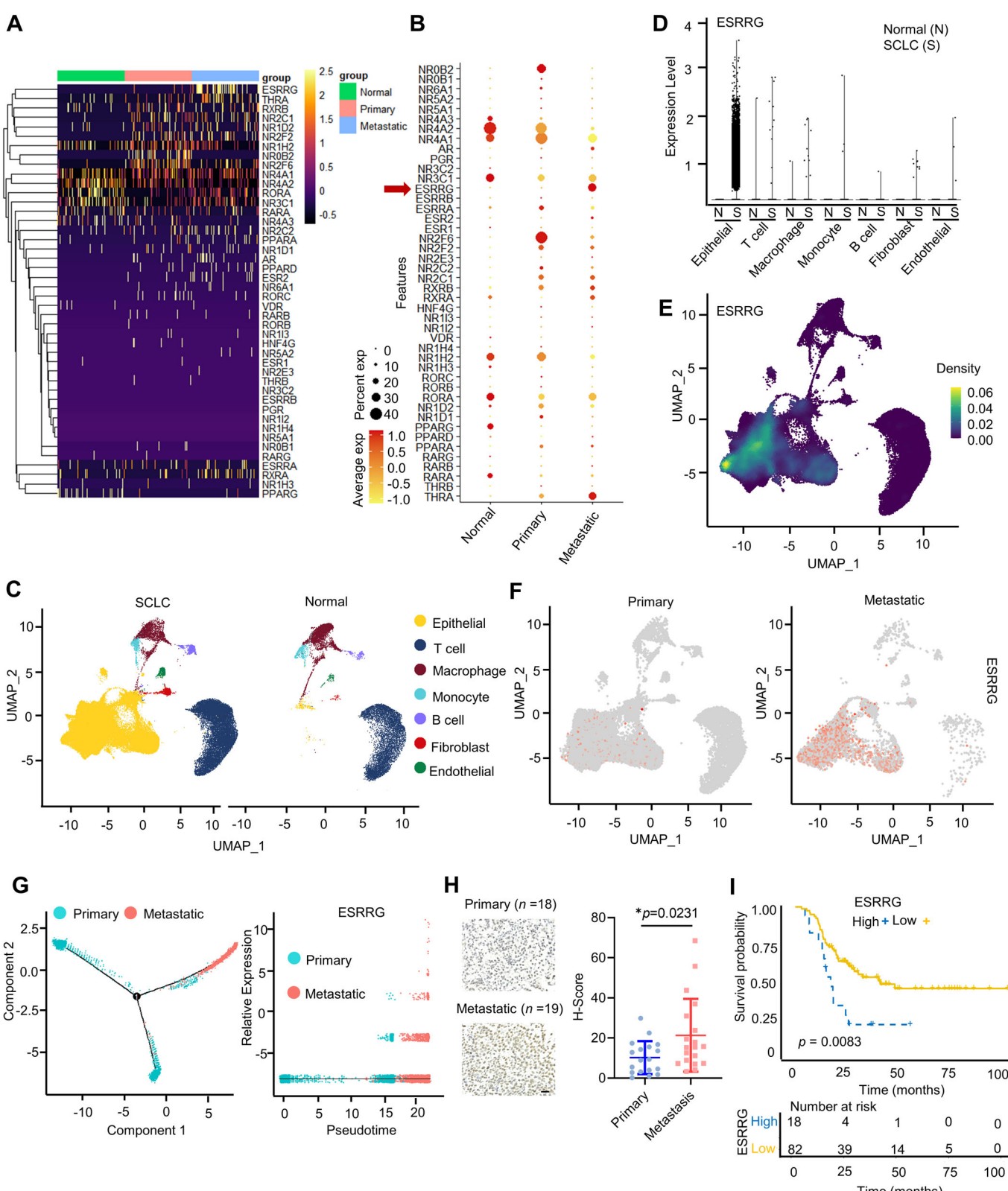

Having demonstrated that ERRγ is crucial for the growth and survival of SCLC cells in vitro, we evaluated the impact of ERRγ inhibition on SCLC tumor growth in vivo. As shown in Fig. 2J, ESRRG knockdown strongly suppressed tumorigenesis and the growth of H446

SCLC xenograft tumors (Appendix Fig. S2K). In parallel, intraperitoneal administration of DN200434 potently inhibited tumor growth in H446 tumor-bearing mice without causing any significant changes in body weight (Fig. 2K; Appendix Fig. S2P). Immunohistochemical

**Figure 1. ESRRG is overexpressed in SCLC and is associated with SCLC metastasis.**

(A–G) Single-cell sequencing analysis of tumor samples from SCLC patients with tumors at different stages (syn23593846, $n = 21$ patients). (A) Heatmap showing the RNA expression levels of the indicated NRs in normal lung tissues, primary tumors and metastatic tumors ($n = 21$ patients). The Single-cell analysis was conducted following previously published protocols (Han et al, 2022). (B) Dot plot showing the mean level of indicated NR expression (dot intensity, red scale) and percent of cells in the population with detected expression (dot size), corresponding to normal lung tissues, primary tumors and metastatic tumors ($n = 21$ patients). (C) UMAP visualization of all cells clustered and colour coded by cell type. (D) Expression level of ESRRG in different cell types from normal lung tissues and SCLC tumors. (E) UMAP visualization of all cells overlaid with the expression of ESRRG. (F) UMAP visualization showing ESRRG expression in cells from primary tumors (left) and metastatic tumors (right). (G) Expression of ESRRG during the cancer cell state transition. (H) Representative images (left) obtained through ERRγ immunohistochemistry and a statistical diagram (right) showing the H-Score of primary SCLC ($n = 18$) and metastatic SCLC ($n = 19$) tumor specimens. Scale bar, 20 μm. (I) The Kaplan–Meier survival curves for SCLC patients stratified into high- and low-ESRRG groups. The $P$ value was calculated using the log-rank test ($n = 18$ for high ESRRG; $n = 82$ for low ESRRG). Data information: Data represent different numbers ($n$) of biological replicates. Data shown in (H) are presented as mean ± s.d. Student's $t$ test is used in (H). Log-rank test is used in (I). *$P < 0.05$. Source data are available online for this figure.

staining of the tumor sections further confirmed that DN200434 effectively inhibited SCLC cell proliferation and induced apoptosis in vivo, as evidenced by Ki-67 staining and cleaved caspase-3 staining, respectively (Fig. 2L; Appendix Fig. S2L–O). These findings suggest that ERRγ is a crucial determinant of SCLC cell survival both in vitro and in vivo.

## ERRγ inhibition suppresses SCLC cell invasion and tumor metastasis both in vitro and in vivo

Given the significant correlation between elevated ESRRG expression and SCLC metastasis (Fig. 1), we investigated whether ERRγ controls SCLC cell invasion and tumor metastasis. As shown in Fig. 3A–D, both pharmacological inhibition and knockdown of ERRγ significantly inhibited migration and invasion by SCLC cells in vitro (Appendix Fig. S3A–D). In contrast, overexpression of ERR promoted invasion by SCLC cells (Fig. 3E; Appendix Fig. S3E).

We then examined the role of ERRγ in SCLC metastasis in vivo. At the time of initial diagnosis, two-thirds of SCLC patients already have metastatic disease, most frequently involving the liver, lymph nodes, brain, and bones. The liver is the predominant site of metastasis in SCLC patients (Ko et al, 2021). Previous studies reported that intrasplenic injection of tumor cells can result in hematogenous metastasis into the liver, and these studies eventually led to the construction of a liver metastasis model (Simons et al, 2020; Li et al, 2019; Jiang et al, 2021; Zhang et al, 2023; Ji et al, 2022). Inspired by this, we generated H1048-Luc-EGFP cells that stably express luciferase, an enzyme that catalyses the oxidation of luciferin to produce bioluminescence. H1048-Luc-EGFP cells were injected into the spleens of the mice, and the bioluminescence intensity was measured using IVIS imaging 3 days later and for the following 4 weeks. We observed a significant reduction in luciferase activity after intraperitoneal injection of DN200434 compared to that in the control group (Fig. 3F). Consistently, the luciferase activity and H&E staining of liver sections demonstrated that DN200434 effectively suppressed SCLC tumor metastasis (Fig. 3G).

To further investigate the role of ERRγ in SCLC metastasis, we established a metastasis model by tail vein injection of tumor cells (Ko et al, 2021; Kuramoto et al, 2012; Sakamoto et al, 2020; Sato et al, 2013; Wang et al, 2020). First, H69AR-Luc-EGFP cells were injected into the tail veins of mice, and bioluminescence intensity was measured using IVIS imaging system on this day and continuing for 2 weeks. Consistent with the results obtained in the liver metastasis model, the luciferase activity showed significantly downregulated in DN200434 treatment group compared to the control group (Fig. 3H,I). H&E staining of lung tissue revealed a

reduction in metastatic foci in mice treated with DN200434 (Fig. 3I). Furthermore, immunohistochemical analysis showed that DN200434 effectively inhibited the expression of metastasis maker genes β-catenin and Vimentin in tumors from the both H1048 and H69AR metastasis models (Appendix Fig. S3F,G). Taken together, these results strongly suggest that ERRγ inhibition can potently suppress SCLC cell invasion and tumor metastasis. Considering this alongside the effect of ERRγ inhibition on SCLC cell survival (Fig. 2), our findings demonstrate that while we cannot exclude the potential influence of ERRγ inhibition-induced suppression of cell survival in these experiments, they do indicate the significant role of ERRγ in SCLC metastasis.

## ERRγ reprograms ECM and cell adhesion signalling

To determine the core transcriptional programs governed by ERRγ in SCLC cells, we conducted RNA-seq analysis of H128 cells transfected with siESRRGs or siCont (Fig. 4A). Gene Ontology (GO) analysis of the transcripts commonly downregulated by the two different ERRγ siRNAs revealed that several ECM signalling pathways, including those associated with the matrisome, ECM regulators, cell adhesion molecules and extracellular matrix organization, were among the top gene signatures (Fig. 4B). These findings were confirmed by Gene Set Enrichment Analysis (GSEA), which indicated that ESRRG knockdown also downregulated the cell adhesion molecule pathway, the focal adhesion pathway, and the ECM-receptor interaction pathway (Fig. 4C). Pathway-focused analysis further demonstrated that the expression of major genes that participate in ECM remodeling pathways, including those encoding ECM remodeling enzymes, adhesion molecules, and ECM components, was significantly downregulated by siESRRGs (Fig. 4D). qRT-PCR and western blotting verified that expression of the key genes involved in the ECM remodeling pathway was strongly inhibited by ESRRG knockdown at both the mRNA and protein levels (Fig. 4E,F). Consistent with this, the ERRγ antagonist DN200434 also inhibited the expression of these genes (Appendix Fig. S4A,B).

To determine whether the nuclear receptor ERRγ directly controls the transcription of ECM-related genes, we first analysed a published ERRγ ChIP-Seq dataset generated from cancer cells (Dunham et al, 2012). We found that ERRγ binds to the promoter or enhancer regions of key ECM-related genes such as MMP9, WNT4, CNTN2 and NECTIN2 (Fig. 4G; Appendix Fig. S4C). Our ChIP-qPCR experiments confirmed that ERRγ directly binds to the target loci of CNTN2, MMP9, NECTIN2 and WNT4, and its antagonist DN200434 significantly reduced ERRγ occupancy of

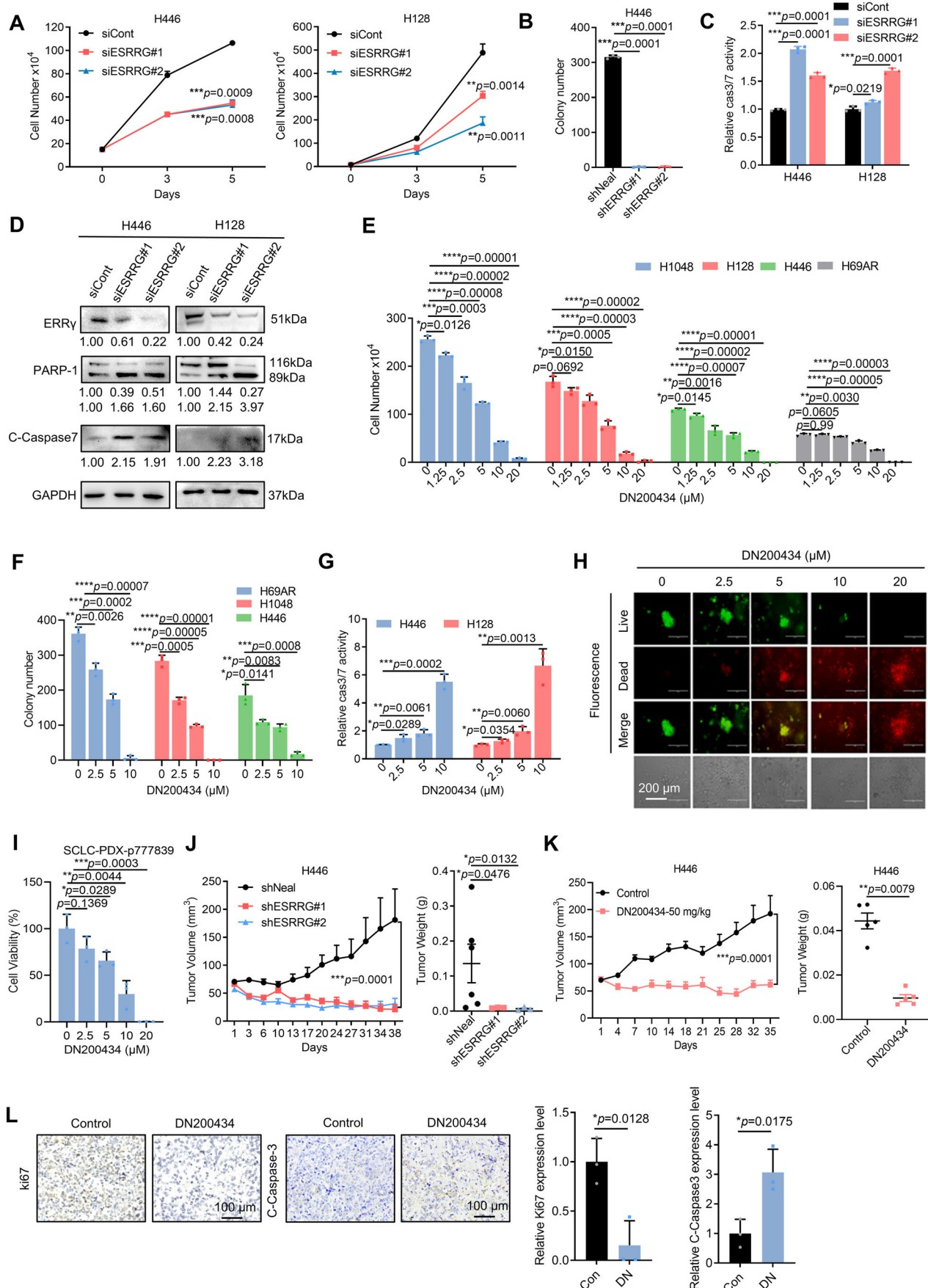

◄ **Figure 2. ERRγ is a major driver of SCLC cell survival and tumor tumorigenesis.**

(A) H446 and H128 cells were transfected with siRNA against ESRRG. The number of cells was counted on Days 3 and 5 ($n = 3$ biological replicates). (B) H446 cells were transfected with shRNA against ESRRG. Fourteen days later, colonies were counted ($n = 3$ biological replicates). (C) Caspase 3/7 activities were measured using a luminescent caspase-Glo 3/7 assay kit with SCLC cells harvested 3 days after infection as described in (A) ($n = 3$ biological replicates). (D) Immunoblotting analysis of apoptosis-related protein levels in SCLC cells treated as described in (A) and cultured for 3 days. (E) SCLC cells were treated with vehicle or DN200434 as indicated. After 96 h, the total number of viable cells was counted ($n = 3$ biological replicates). (F) SCLC cells were treated with vehicle or with the indicated concentrations of DN200434 for 14 days, after which colony formation was assessed ($n = 3$ biological replicates). (G) Effect of DN200434 on apoptosis. Cells were treated with vehicle or with the indicated concentrations of DN200434 for 72 h; apoptosis was then measured using a luminescent caspase-Glo 3/7 assay kit ($n = 3$ biological replicates). (H) PDX-derived organoids were treated with DMSO or with the indicated concentrations of DN200434. Representative images were obtained using a fluorescence microscope (top three rows) or a standard light microscope (bottom row) ($n = 3$ biological replicates). Scale bar, 200 μm. (I) Statistical analysis of the cell viability of the organoid-derived cells as measured using Cell-Titer Glo 3D ($n = 3$ biological replicates). (J) H446-shNeal, H446-shESRRG#1 and H446-shESRRG#2 cells were injected subcutaneously into nude mice (shNeal, $n = 9$ mice per group; shESRRG#1, shESRRG#2, $n = 7$ mice per group). Mean tumor volume ± s.e.m. (left), mean tumor weight ± s.e.m. (right), *$P < 0.05$, **$P < 0.01$. (K) Mice bearing H446 tumors were treated with vehicle or with 50 mg/kg DN200434 ($n = 5$ mice per group). Mean tumor volume ± s.e.m. (left), mean tumor weight ± s.e.m. (right), **$P < 0.01$. (L) Immunoblotting of H446 tumors after treatment with vehicle or DN200434, as shown in (K). The images on the left show the results of immunohistochemical staining with an antibody against Ki67 and an antibody against Cleaved Caspase-3; a statistical analysis of the tumor sections is shown on the right. Scale bar, 100 μm ($n = 3$ biological replicates). Data information: Data represent different numbers ($n$) of biological replicates. Data shown in (A–C, E–I, L) are presented as mean ± s.d. Data shown in (J, K) are presented as mean ± s.e.m. Student's $t$ test is used in (A–C, E–L). ns not significant. *$P < 0.05$, **$P < 0.01$, ***$P < 0.001$, ****$P < 0.0001$. Source data are available online for this figure.

these sites (Fig. 4H). Consistent with the loss of ERRγ occupancy, our ChIP-Seq and ChIP-qPCR data demonstrated that the transcriptional activation-linked histone mark H3K27ac at the promoters and/or enhancers of ECM-related genes such as CNTN2, MMP9, NECTIN2 and WNT4 was significantly reduced by DN200434 (Fig. 4G,I; Appendix Fig. S4C). Collectively, these findings indicate that ERRγ plays a crucial role in activating signaling related to ECM remodeling by directly regulating the transcription of ECM-related genes in SCLC cells.

## ERRγ is a major regulator of ECM remodeling and cell adhesion

The prominent role of ERRγ in controlling the ECM program prompted us to examine whether ERRγ can remodel the extracellular matrix in a way that facilitates metastasis and invasion in SCLC. The ECM is a dynamic structure that undergoes continuous remodeling by proteases produced by a variety of cells in the TME (Winkler et al, 2020). Dysregulation of ECM remodeling by cancer cells contributes to the creation of a microenvironment that promotes tumor growth and metastasis (Yuan et al, 2023). Increased deposition of fibrillar collagen is a common tumorigenic alteration of ECM homeostasis (Winkler et al, 2020). To characterize the collagen produced by tumor cells, we used the fluorescent collagen probe CNA35-mCherry (Aper et al, 2014), which binds specifically to collagen. Genetic or pharmacological inhibition of ERRγ in SCLC cells effectively decreased collagen deposition in the ECM as assessed using the collagen probe CNA35-mCherry, qRT-PCR and western blotting for COL6A1 (Figs. 5A,B,G and 4E,F; Appendix Fig. S4A,B). Moreover, treatment with the ERRγ antagonist DN200434 also significantly inhibited collagen deposition in H446 tumor sections in vivo (Fig. 5C,D). Consistent with the suppression of adhesion molecules expression by ESRRG (Fig. 4D), DN200434 potently attenuated the Matrigel adhesion ability of SCLC cells (Fig. 5I). Adhesion molecules facilitate bidirectional signalling between ECM components and the actin cytoskeleton within cells, and reorganization of the actin cytoskeleton mediates cellular processes such as cell migration (Schmidt and Friedl, 2010; Sun et al, 2016). We measured the effect of ERRγ on the cytoskeleton using phalloidin staining; the results showed that ERRγ inhibition remodeled the

cytoskeleton and affected the arrangement of the microfilaments produced by SCLC cells (Fig. 5E,F,H). These findings suggest that ERRγ regulates collagen production, cell-matrix adhesion, and microfilament production, thereby eventually establishing a microenvironment that facilitates SCLC metastasis.

## ERRγ antagonists sensitize chemoresistant SCLC tumors to chemotherapy

The extracellular matrix (ECM) plays a crucial role in chemoresistance in SCLC (Sethi et al, 1999). To gain functional understanding of SCLC chemoresistance associated with ERRγ and its regulation on ECM, we first examined the expression of ERRγ and ECM signature genes as revealed by scRNA-seq data from naïve and chemoresistant SCLC tumors. ERRγ expression and the ECM activity signature were strongly elevated in cancer cells obtained from patients who showed resistance to chemotherapy and/or immunotherapy. The expression of these genes was positively correlated in chemoresistant samples, indicating a potential role of ERRγ in SCLC drug resistance (Fig. 6A–C; Appendix Fig. S5A–C). We then examined whether the ERRγ antagonist DN200434 could mitigate SCLC chemoresistance. Using a chemoresistant PDO culture system, we discovered that treatment with a combination of EP and DN200434 had greater efficacy in inhibiting PDO growth than did single-agent therapy (Fig. 6D,E).

Next, we designed two sets of in vivo experiments in which we examined the therapeutic potential of combined EP and DN200434 chemoresistant SCLC. First, we generated xenograft mouse models by intravenously injecting H69AR-Luc-EGFP cells into the tail veins of mice; we then treated the mice with DN200434 and/or EP daily for 4 weeks. The results showed that the combination treatment synergistically inhibited H69AR-derived tumor growth and metastasis (Fig. 6F). Moreover, examination of multiorgan-site metastasis revealed that the combined treatment almost completely eliminated metastasis to the brain, lung, liver and kidney (Fig. 6G) as shown in a previous section (Fig. 3G,I). To provide data that are more predictive of SCLC patients' therapeutic outcomes, we developed a patient-derived chemoresistant SCLC xenograft (PDX) model and assessed the efficacy of DN200434 alone and that of DN200434 in combination with EP in inhibiting PDX tumor growth. DN200434 alone significantly inhibited PDX tumor growth and treatment with

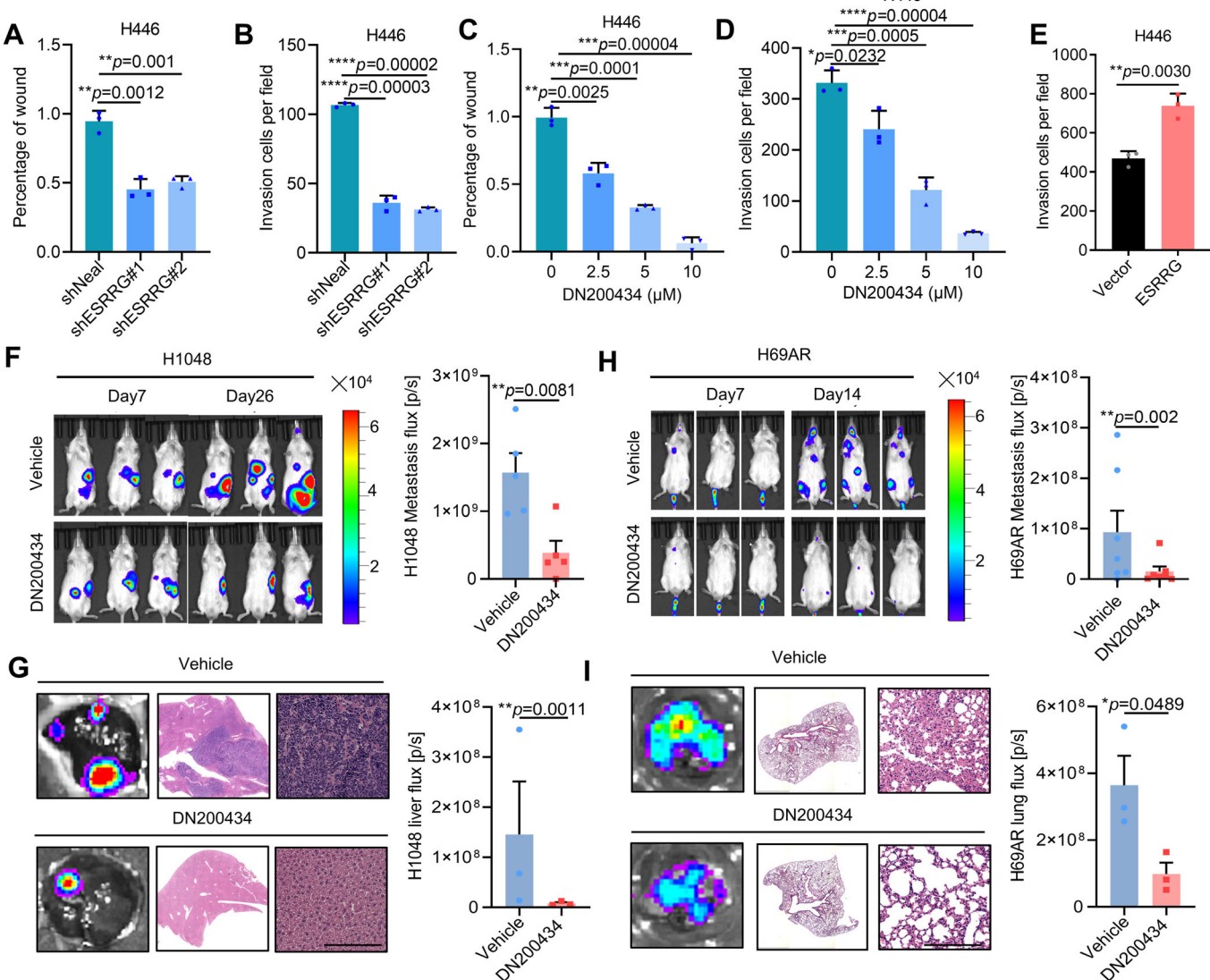

**Figure 3. ERRγ promotes SCLC cell invasion and metastasis.**

(A) Statistical analysis of the results obtained in the wound healing assay in H446 cells with or without ESRRG depletion ($n = 3$ biological replicates). (B) Cell numbers of transwell assays from the indicated groups with or without ESRRG knockdown in H446 cells ($n = 3$ biological replicates). (C) Statistical analysis of the results obtained in the wound healing assay in H446 cells treated with vehicle or DN200434 ($n = 3$ biological replicates). (D) Cell numbers of transwell assays from the indicated groups treated with vehicle or DN200434 in H446 cells ($n = 3$ biological replicates). (E) Cell numbers of transwell assays from wild-type and ERRγ overexpressed H446 cells ($n = 3$ biological replicates). (F) H1048-Luc-EGFP cells were transplanted into the spleens of male NOD-SCID mice through transdermal injection to establish a model of SCLC liver metastasis. The mice were then treated with DN200434 (25 mg/kg/day, i.p.) or vehicle ($n = 5$ mice per group). Representative luminescence images (left) and a diagram showing a statistical analysis of the average luminescence intensity in the mice (right) are shown. The data shown are the mean ± s.e.m. $P = 0.0081$. (G) Representative luminescence images (left) and H&E staining (middle) of liver sections from the mice shown in (F) on the final day. Statistical analysis (right) of the average luminescence intensity in liver sections of mice bearing H1048-Luc-EGFP cells. The data shown are the mean ± s.e.m. $n = 3$ per group, $P = 0.0011$. Scale bar, 125 μm. (H) H69AR-Luc-EGFP cells were injected into the tail veins of mice to establish a metastasis model. The injected mice were then treated with DN200434 (25 mg/kg/day, i.p.) or with vehicle ($n = 6$ mice per group). Tumour growth was monitored by bioluminescence imaging. Representative luminescence images (left) and a statistical analysis (right) of the average luminescence intensity in the mice are shown. The data are presented as the mean ± s.e.m. $P = 0.002$. (I) Representative luminescence images (left) and H&E staining (middle) of lung sections of the mice shown in (H) on the final day. Statistical analysis (right) of the average luminescence intensity observed in lung sections of mice bearing H69AR-Luc-EGFP cells. The data shown are the mean ± s.e.m. $n = 3$ per group, $p = 0.0489$. Scale bar, 125 μm. Data information: Data represent different numbers ($n$) of biological replicates. Data shown in (A–E) are presented as mean ± s.d. Data shown in (F–I) are presented as mean ± s.e.m. Student's $t$ test is used in (A–I). *$P < 0.05$, **$P < 0.01$, ***$P < 0.001$, ****$P < 0.0001$. Source data are available online for this figure.

DN200434 and EP together synergistically inhibited tumor growth (Fig. 6H; Appendix Fig. S5D). IHC staining of sections of the tumor tissue further confirmed the efficacy of the combination treatment, as it showed that it decreased proliferation and increased cell apoptosis

(Fig. 6I; Appendix Fig. S5E–G). Collectively, our data indicate that treatment with an ERRγ antagonist alone or in combination with chemotherapy can represent a novel strategy for the effective management of advanced SCLC.

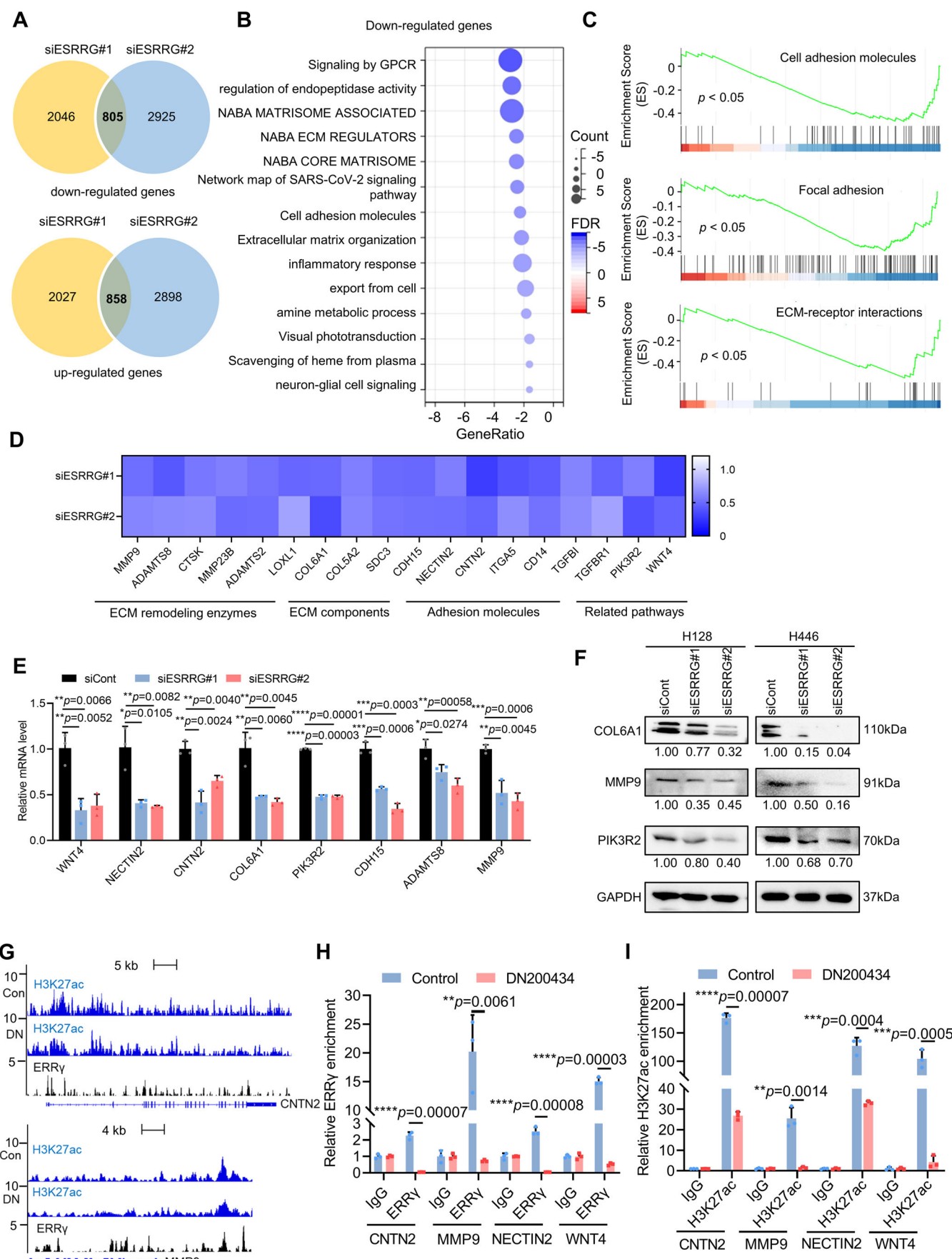

**Figure 4.   ERRγ inhibition reprograms ECM signalling.**

(A) A Venn diagram showing the number of genes whose expression was significantly (1.2-fold) downregulated (top) and upregulated (bottom) was generated via RNA-seq of H128 cells with or without ESRRG knockdown. (B) Gene Ontology analysis of the 805 genes downregulated by siESRRG#1 and siESRRG#2 shown in (A). (C) GSEA of the expression of cell adhesion molecules and focal adhesion and ECM-receptor interaction signatures in H128 cells transfected with siESRRG compared to those in H128 cells transfected with vehicle. (D) Heatmap displaying the fold changes in gene expression detected by RNA-seq in H128 cells transfected with siESRRG compared to those transfected with vehicle. (E) qRT-PCR analysis of the indicated genes in H128 cells transfected with siESRRG or vehicle ($n = 3$ biological replicates). (F) Immunoblotting of the indicated proteins in H446 and H128 cells treated with vehicle or ESRRG siRNAs. (G) ChIP-Seq profiles of H3K27ac and public ChIP-Seq profiles of ERRγ binding around the centre of peak regions on genes involved in the ECM remodeling pathway. (H) ChIP-qPCR analysis of the presence of ERRγ at the indicated gene promoters and/or enhancers in H128 cells treated with vehicle or with 5 μM DN200434 for 48 h ($n = 3$ biological replicates). (I) ChIP-qPCR analysis of the presence of H3K27ac at the indicated gene promoters and/or enhancers in H128 cells treated with vehicle or with 5 μM DN200434 for 48 h ($n = 3$ biological replicates). Data information: Data represent different numbers ($n$) of biological replicates. Data shown in (E, H, I) are presented as mean ± s.d. Student's $t$ test is used in (E, H, I). *$P < 0.05$, **$P < 0.01$, ***$P < 0.001$, and ****$P < 0.0001$. Source data are available online for this figure.

## Discussion

Metastasis remains the leading cause of death in patients with SCLC, largely due to a lack of effective therapeutic targets and treatment options. Here, we provide evidence that supports the use of ERRγ as a promising therapeutic target for SCLC. We discovered that ERRγ is highly overexpressed in SCLC tumors, especially at metastatic sites. ERRγ functions as a key determinant of ECM-related gene expression in SCLC cells. Genetic and pharmacological inhibition of ERRγ markedly reduced collagen production, cell-matrix adhesion, and microfilament production, and substantially suppressed SCLC tumor cell growth and metastasis both in vitro and in vivo. Furthermore, an ERRγ antagonist significantly increased SCLC sensitivity to chemotherapy in multiple preclinical models. These findings indicate that ERRγ plays a crucial role in the development and metastasis of SCLC. Moreover, considering the favorable druggability of ERRγ, it highlights ERRγ as a promising therapeutic target in this deadly disease.

Targeted therapy has achieved significant breakthroughs over the past three decades for various cancer types. Over 30 targeted drugs, including EGFR inhibitors and ALK inhibitors, are extensively used in the clinical treatment of non-small cell lung cancer (NSCLC), and use of these drugs has resulted in a substantial prolongation of patient survival. However, no targeted drugs have been approved for the treatment of SCLC, and all NSCLC-targeted therapeutics have proven unsuccessful in clinical trials for SCLC therapy. Recently, improved molecular understanding has brought new promising opportunities for SCLC treatment, and several inhibitors targeting MYC (Mollaoglu et al, 2017), DLL3 (Jaspers et al, 2023), EZH2 (Gardner et al, 2017), LSD1 (Chen et al, 2022a) and CDK7 (Zhang et al, 2020) have entered clinical trials. We and others previously reported that several orphan receptors, including RAR-related orphan receptor gamma (RORγ) and Nur77, are involved in SCLC growth and survival (Chen et al, 2022b; Sanada et al, 2023; Payapilly et al, 2021). However, little is known about which druggable factor(s) drive metastasis in the context of SCLC. In this work, we provide the first evidence that druggable ERRγ not only plays a crucial role in cell growth but also promotes cell metastasis and invasion in SCLC. Genetic and pharmacological inhibition of ERRγ significantly inhibited cell survival and invasion. An ERRγ antagonist potently inhibited the growth and metastasis of multiple different SCLC xenograft tumors in mice. Given the primary role of chemotherapy resistance in SCLC treatment failure, we developed a

cell-based and patient-derived chemoresistant SCLC xenograft models and used them to evaluate the efficacy of combination treatment with an ERRγ antagonist and chemotherapy for inhibiting tumor growth and metastasis. The results we obtained highlight the strong synergistic effect of the ERRγ antagonist DN200434 and chemotherapeutic agents (EP) on SCLC tumor growth and metastasis. Thus, therapies targeting ERRγ could have broad clinical utility in SCLC.

ERRγ belongs to the nuclear receptor superfamily of transcription factors and is directly linked to gene expression. Our data show that inhibition of ERRγ reprograms gene expression patterns in SCLC cells. Multiple ECM signalling pathways were among the top gene signatures whose expression was altered by ERRγ inhibition. SCLC is surrounded by an extensive stroma of ECM, and high levels of expression of ECM genes correlate with poor patient prognosis (Hodkinson et al, 2007). ECM not only protects SCLC cells against chemotherapy-induced cell death but also facilitates tumor invasion, metastasis and angiogenesis (Hodkinson et al, 2007; Yuan et al, 2023; Sethi et al, 1999; Gilkes et al, 2014). Our data show that ERRγ exerts diverse impacts on the ECM remodeling process in SCLC by directly regulating the expression of key genes involved in ECM remodeling and cell adhesion, including MMP9, CNTN2, NECTIN2, and WNT4. Matrix metallopeptidase 9 (MMP9), a member of the MMP family, is a major regulator of extracellular matrix turnover and thereby facilitates tumor metastasis (Gilkes et al, 2014). Contactin 2 (CNTN2) is a cell adhesion molecule that regulates the receptor tyrosine kinase (RTK)/Ras/extracellular signal-regulated kinase (ERK) MAPK signalling pathway (Kastriti et al, 2019; Yan and Jiang, 2016). Nectin cell adhesion molecule 2 (NECTIN2), an upstream target of cytoskeletal regulation via SRC signaling, is an adhesion molecule that has been reported to play a role in tumor growth, metastasis and angiogenesis (Bekes et al, 2019; Son et al, 2016). Phosphoinositide-3-kinase regulatory subunit 2 (PIK3R2), a subunit of PI3K, plays an important role in cell growth and proliferation (Liu et al, 2022). WNT4 is recognized as an epithelial to mesenchymal transition (EMT)-related transcription factor that functions as an autocrine and paracrine signalling protein within the ECM (Afrin et al, 2022). Inhibition of ERRγ markedly reduced collagen production, cell-matrix adhesion, and microfilament production, and eventually blocked SCLC cell invasion and tumor metastasis. Our data indicate that the ECM activity signature is significantly increased in resistant SCLC cells, potentially shielding them from chemotherapy-induced cell death. In contrast, an ERRγ antagonist notably enhanced the sensitivity of resistant SCLC to chemotherapy. These results show that ERRγ systematically regulates the interactions between cancer cells and the

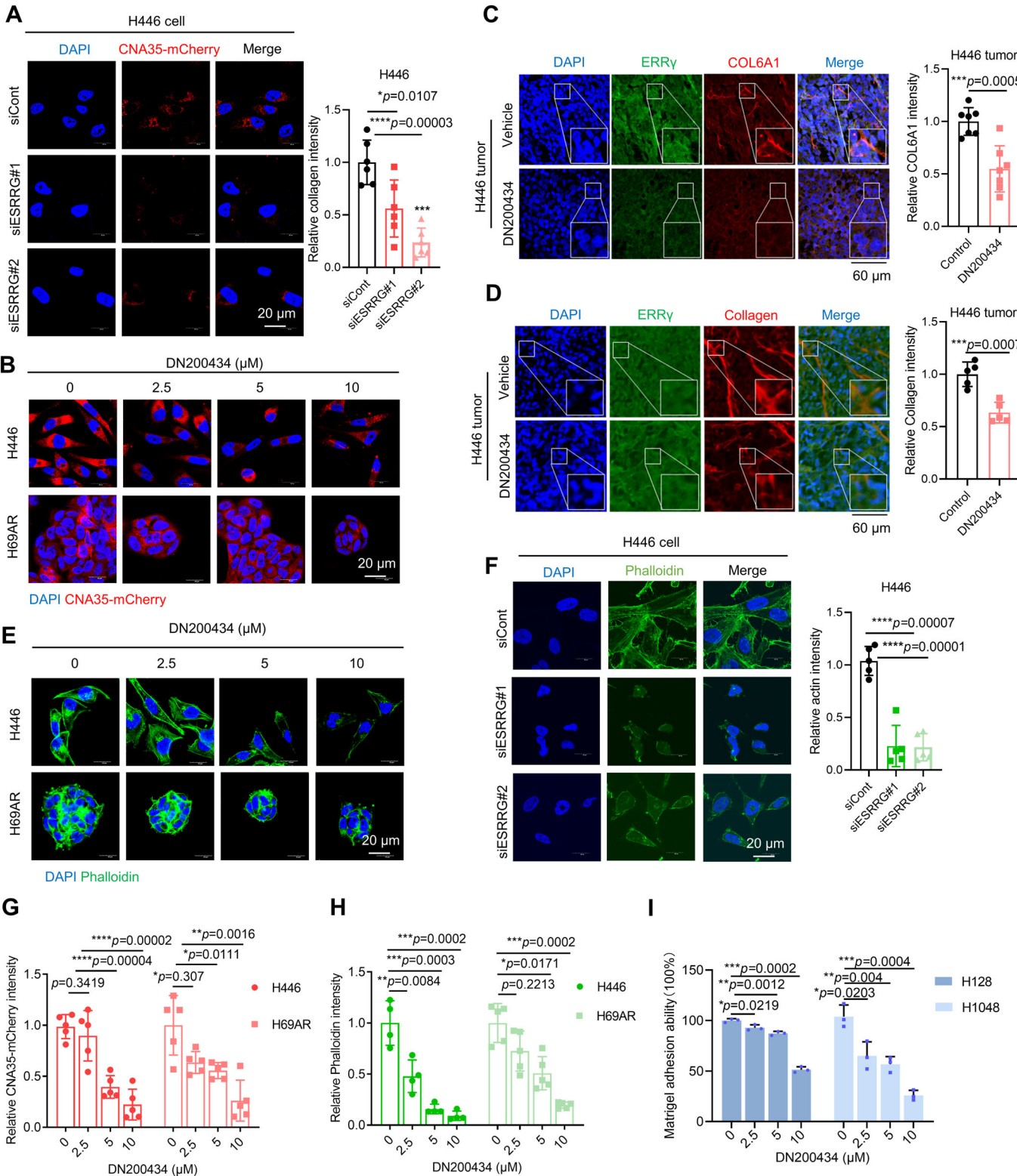

matrix environment, thereby facilitating SCLC tumor development
and metastasis.

ERRγ is a member of an orphan nuclear receptor family, whose
members possess the common structural features of nuclear receptors
and contain a ligand-binding domain, making it amenable to the

development of drugs targeted to act on small molecules. Multiple
ERRγ antagonists, including GSK5182 and DN200434, have been
developed. DN200434, which was utilized in this study, exhibits
promising selective binding affinity for ERRγ and high activity when
applied to cells (Kumar et al, 2022). However, the in vivo antitumor

**Figure 5. ERRγ is a major regulator of ECM remodeling and cell adhesion.**

(A) Representative images and quantification of collagen produced by H446 cells treated with siESRRG or siCont (n = 3 biological replicates). Scale bar, 20 μm. (B) Representative images showing collagen produced by H446 or H69AR cells treated with DMSO or with 2.5 μM, 5 μM and 10 μM DN200434 (n = 3 biological replicates). Scale bar, 20 μm. (C) Immunofluorescence staining and quantification of ERRγ (green) and COL6A1 (red) in H446 tumor sections treated with or without 50 mg/kg DN200434 (n = 3 biological replicates). Scale bar, 60 μm. (D) Immunofluorescence staining and quantification of ERRγ (green) and collagen (red) in H446 tumor sections treated with or without 50 mg/kg DN200434 (n = 3 biological replicates). Scale bar, 60 μm. (E) Representative images and quantification of the F-actin cytoskeleton labeled with iFluorTM 488 phalloidin in H446 cells treated with siESRRG or siCont (n = 3 biological replicates). Scale bar, 20 μm. (F) Representative images of the F-actin cytoskeleton labelled with iFluorTM 488 phalloidin in H446 and H69AR cells treated with DMSO or with 2.5 μM, 5 μM and 10 μM DN200434 (n = 3 biological replicates). Scale bar, 20 μm. (G) Quantification of collagen produced by H446 and H69AR cells treated with DMSO or with 2.5 μM, 5 μM and 10 μM DN200434 (n = 3 biological replicates). (H) Quantification of the F-actin cytoskeleton in H446 and H69AR cells treated with DMSO or with 2.5 μM, 5 μM and 10 μM DN200434 and labelled with iFluorTM 488 phalloidin (n = 3 biological replicates). (I) Matrigel adhesion ability of SCLC cells treated with DN200434 or vehicle (n = 3 biological replicates). Data information: Data represent different numbers (n) of biological replicates. Data shown in (A, C, D, F–I) are presented as mean ± s.d. Student's t test is used in (A, C, D, F–I). *P < 0.05, **P < 0.01, ***P < 0.001, ****P < 0.0001. Source data are available online for this figure.

efficacy of DN200434 still requires improvement. To enhance in vivo antitumor activity, novel strategies should be employed to develop more potent ERRγ antagonists. Proteolysis targeting chimaera (PROTAC) technology, a powerful approach, provides a more complete and sustained inactivation of target protein signalling by degrading the protein of interest (POI). Recently, several nuclear receptor PROTAC degraders, including ARV-110 and ARV-471, have begun to be tested in clinical trials, and some of these show promising results (Qi et al, 2021). Therefore, the development of ERRγ PROTAC degraders could constitute a novel strategy for the development of ERRγ-targeted drugs for SCLC treatment.

In summary, effective therapeutic strategies for patients with metastatic and chemoresistant SCLC are sorely needed. Our study revealed that ERRγ is a crucial factor in ECM remodeling and a promising therapeutic target for advanced SCLC. Given the druggable nature of ERRγ and the availability of multiple potent ERRγ antagonists (Chao et al, 2006; Singh et al, 2019), our findings are likely to contribute to the development of new therapeutic strategies for advanced SCLC.

# Methods

## Patient samples and ethical statement

Human SCLC surgical specimens (from 37 patients; 18 primary SCLC samples and 19 metastatic SCLC samples) were collected from The Third Affiliated Hospital of Sun Yat-Sen University and The First People's Hospital of Foshan. Human SCLC PDXs were obtained from Guangdong Provincial People's Hospital and West China Hospital of Sichuan University. This study was approved by the Hospital Medical Ethics Committee (Approved number: RG2023-240-02) and informed consent was obtained from all participating patients or their guardians. The experiments performed on human-derived tumor samples conformed to the principles set out in the WMA Declaration of Helsinki and the Department of Health and Human Services Belmont Report.

## Cell culture

The human small cell lung cancer cell lines, H446, H1048, H69AR, H128 and HEK293T were obtained from American Type Culture Collection (ATCC). H446, H128 and H69AR cells were cultured in RPMI1640 medium. H1048 cells and HEK239T cells were cultured

in DMEM. All cell culture media were supplemented with 10% foetal bovine serum (FBS) and 1% penicillin/streptomycin (Gibco). All cells were cultured at 37 °C in 5% CO$_2$ incubators. The SCLC cell lines were recently authenticated by using STR profiling. Cell lines were regularly tested being negative for mycoplasma. The chemicals used in this study are listed in Appendix Table S1.

## Colony formation

A total of 3000–5000 cells were seeded in six-well plates and incubated for approximately 14 days with the indicated treatments. The culture medium was changed every 3 days. After 14 days in culture, the cells were washed with PBS and fixed with 4% paraformaldehyde for 20 min. The cells were then washed again with PBS, stained with before crystal violet staining for 20 min, and rinsed three times with PBS before cell number counting.

## Cell growth and caspase-3/7 activity assays

For the cell growth assay, $1.5 \times 10^5$ cells were seeded in 6-well plates with the indicated treatments and incubated for 96 h. The culture medium was changed every 2 days. The total number of viable cells was counted using a Coulter cell counter.

For the caspase-3/7 activity assay, $1.5 \times 10^5$ cells were seeded in six-well plates and incubated for 72 h with the indicated treatments. Caspase-3/7 activity was measured using a luminescent caspase-Glo 3/7 assay kit (Promega Corporation, Madison, WI, USA), according to the manufacturer's instructions. The total protein concentration was measured and used to normalize the results.

## Cell migration assays

For cell migration assays, $6 \times 10^5$ cells were seeded in six-well plates, and the surface of the plates was scratched on the following day. The cells were incubated for 72 h with the indicated treatments. Images were collected every 24 h. Scratch confluency was quantified using ImageJ software.

## Transwell assays

For transwell assays, $3 \times 10^4$ cells were resuspended in serum-free medium and inoculated onto transwell filters with 8-mm pores coated with matrix gel. The filters were placed in 24-well plates containing medium supplemented with 10% FBS. The cells were incubated for 48 h with the indicated treatments; they were then

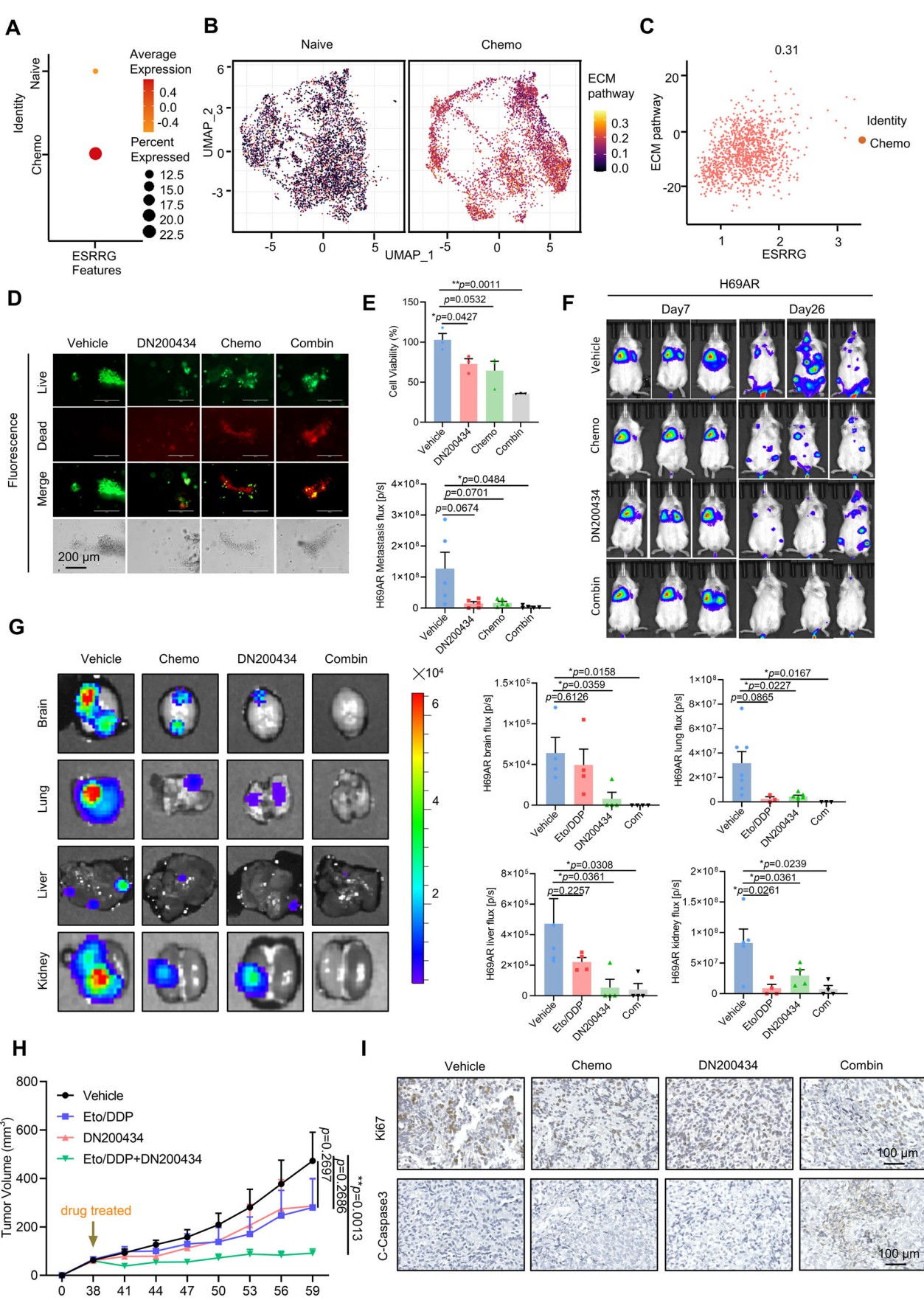

**Figure 6. ERRγ inhibition sensitizes SCLC tumor to chemotherapy.**

(A) Dot plot showing the mean level of ESRRG expression (dot intensity, red scale) and the percentage of cells in the population with detected expression (dot size) in treatment-naïve samples and chemotherapy-treated samples. (B) UMAP visualization showing the AUC score of the ECM pathway in all cells from treatment-naïve samples (left) and chemotherapy-treated samples (right). (C) Scatter plot showing the correlation between the relative expression levels of ESRRG and members of the ECM pathway in chemotherapy-treated SCLC samples. (D) PDX-derived organoids were treated with DMSO or with the indicated treatment. Representative images (left) were obtained using a fluorescence microscope (top three rows) or a standard light microscope (bottom row) ($n = 3$ biological replicates). Scale bar, 200 μm. (E) The viability of the organoids described in (D) above was measured using CellTiter-Glo 3D ($n = 3$ biological replicates). (F) H69AR-Luc-EGFP cells were injected into the tail veins of mice to establish a metastasis model. The mice were then treated with vehicle, DN200434 (10 mg/kg/day, i.p.) or Chemo (Day 1, DDP 2.5 mg/kg, i.p.; Days 1–3, Eto 4 mg/kg, i.p.) alone or in combination ($n = 5$ mice per group). One week was considered one cycle. Tumor growth was monitored by bioluminescence imaging. Representative luminescence images (right) and a statistical analysis (left) of the average luminescence intensity in the mice. The data are presented as the mean ± s.e.m. $P$ values are shown in the panels. (G) Representative luminescence images (left) and a statistical analysis (right) of the average luminescence intensity in brain, lung, liver and kidney sections of the mice described in (F) above on the final day. The data are presented as the mean ± s.e.m. $P$ values are shown in the panels ($n = 4$ mice per group). (H) Mice bearing LN140 PDX tumors were treated with vehicle, DN200434 (10 mg/kg/day, i.p.) or Chemo (Day 1, DDP 2.5 mg/kg, i.p.; Days 1–3, Eto 4 mg/kg, i.p.) alone or in combination ($n = 6$ mice per group), 1 week was considered one cycle. The tumor volumes are shown. The data are presented as the mean ± s.e.m. $P$ values are shown in the panels. (I) Representative Ki67 immunostaining and C-Caspase3 immunostaining of tumor sections shown in (H) above on the final day ($n = 3$ biological replicates). Scale bar, 100 μm. Data information: Data represent different numbers ($n$) of biological replicates. Data shown in (D, E) are presented as mean ± s.d. Data shown in (F–H) are presented as mean ± s.e.m. Student's $t$ test is used in (D–H). *$P < 0.05$, **$P < 0.01$. Source data are available online for this figure.

washed three times with PBS and fixed with 4% paraformaldehyde for 20 min. A cotton swab was used to scrape the fixed cells off the tops of the filters. The cells on the bottom sides of the filters were then stained with crystal violet for 20 min, rinsed with PBS, imaged, and counted.

## qPT-PCR and western blotting analysis

Total RNA was extracted from cells in six-well plates. cDNA was prepared, amplified, and measured in the presence of SYBR Green according to the manufacturer's protocol. PCR was performed on a Bio-Rad CFX96™ (Bio-Rad, San Diego, CA, USA). The sequences of the primers used in the qRT-PCR analysis are listed in Appendix Table S2.

Cell lysates were analysed by immunoblotting with antibodies against ERRγ and other indicated proteins. The antibodies used are shown in Appendix Table S3.

## ChIP-qPCR and H3K27ac ChIP-seq

H128 cells were treated with DN200434 for 48 h. Paraformaldehyde was used for crosslinking, and crosslinking was terminated by the addition of glycine. The cells were then washed in PBS, resuspended in lysis buffer (50 mM HEPES, pH 8.0; 140 mM NaCl; 1 mM EDTA; 10% glycerol; 0.5% NP-40; 0.25% Triton X-100), centrifuged, and resuspended in shearing buffer (10 mM Tris-HCl, pH 8.0; 1 mM EDTA, pH 8.0; 0.1% SDS). A Covaris E220 was used to sonicate the cells according to the manufacturer's instructions. Chromatin fragments were precipitated using the indicated antibodies and Protein G beads. The purified ChIP DNA was used in ChIP-qPCR analysis and library generation. H3K27ac ChIP-seq products were prepared according to our previous work (Cai et al, 2019). The primers used are shown in Appendix Table S4.

## siRNA transfection and lentivirus infection

Dharmafectin#1 (Dharmacon) and Opti-MEM (Invitrogen) were used for siRNA transfection according to the manufacturer's instructions. The efficiency of siRNA knockdown was determined by qPT-PCR and western blotting. Lentiviral production was performed in 293T cells as described in our previous work (Wang et al, 2016). The efficiency of lentivirus infection was determined by western blotting. The sequences

of the siRNAs and primers used for the shRNAs are shown in Appendix Tables S5 and Table S6.

## RNA-seq and analysis

H128 cells were transfected with vehicle or ESRRG siRNA for 48 h prior to RNA extraction. Validation of sequence libraries was performed using the MGISEQ2000 SE50 system (BGI Tech, Wuhan, China). In brief, sequence reads were aligned to the reference human genome assembly (GRCh37/hg19) using BWA and Bowtie 2. Gene Set Enrichment Analysis (GSEA V.3.0) was used to rank genes based on log2 fold changes in the expression of the shrunken limma genes. Genes that showed changes in expression exceeding ≥1.2-fold were subjected to clustering using the k-means clustering algorithm within Cluster 63.

## Immunofluorescence (IF) and immunohistochemistry (IHC)

For the IF assays, $5 \times 10^4$ cells were seeded in confocal dishes and subjected to the indicated treatments. After incubation for 48–96 h, the cells were gently washed with PBS and fixed with 4% paraformaldehyde for 10 min. The cells in the dishes were then washed three times with PBS. Triton X-100 (3%) was used for permeabilization. The cells were incubated with 5% normal goat serum for 30 min to block nonspecific antibody binding and then incubated with the indicated antibodies overnight at 4 °C. On the following day, the cells were incubated with the secondary antibody for 1 h at room temperature in the dark. After staining of the cell nuclei with DAPI, images were captured using an OLYMPUS FV3000 confocal laser scanning microscope (Zeiss, Germany). The staining was quantified using ImageJ software.

For the IHC assay, the sections were placed in an oven at 65 °C and then dewaxed using a gradient of xylene and ethanol. After epitope repair, endogenous peroxidase activity was blocked using 3% $H_2O_2$. Normal goat serum (5%) was used to block nonspecific binding of the antibodies. The sections were then incubated with specific antibodies overnight at 4 °C, the primary antibody was removed, and the sections were incubated with the secondary antibody for 1 h at room temperature. After DAB staining, the cell nuclei were stained with Hematoxylin. Images were captured using

an EVOS M7000 imaging system (Thermo Fisher Scientific, USA). The staining was quantified using ImageJ software.

## PDO culture and viability assays

When the tumor size of the PDX xenografts reached approximately 500 mm$^3$, as described in our previous work, organoids were generated from the xenografts (Cai et al, 2019). Briefly, the tumor was trimmed into small pieces, and the pieces were transferred to 50 mL conical tubes, containing 1 mg/mL collagenase IV (Sigma) in serum-free DMEM/F-12 medium (Gibco) and incubated for 1 h at 37 °C.

To measure organoid viability, 300–500 organoids in 5 μL of Matrigel were seeded in 96-well plates. The organoids were cultured in a humidified atmosphere of 5% $CO_2$ at 37 °C. The culture medium was DMEM/F-12 medium containing primocin (50 mg/mL), Y-27632 (5 mM), neuregulin (5 nM), R-Spondin (250 ng/mL), A83-01 (500 nM), FGF (20 ng/mL), HEPES (10 mM), EGF (10 ng/mL), SB202190 (500 nM), nicotinamide (5 mM), B27 supplement (1×), FGF2 (5 ng/mL), and N-acetylcysteine with glutamine/streptomycin/penicillin (100 mg/mL). After the organoids had been cultured for 4 days, the indicated compounds (DN200434 and chemotherapeutic drugs, alone or in combination) were serially diluted in 100 μL of medium and added to the cells. The medium was aspirated, and 100 μL of live/ dead reagents (Thermo Fisher Scientific) was added to the wells. After 30 min at room temperature, images of calcein AM fluorescence (494/517 nm) were captured and used to identify the live cells by fluorescence microscopy; ethidium bromide homodimer-1 (528/617 nm) was used to identify the dead cells. Four days later, the viability of the cells was measured using Cell-Titer Glo reagents (Promega). The results are shown as percentages of the viability of vehicle-treated cells, which was set at 100%.

## Animal experiments

All animal experiments were approved by the Institutional Animal Care and Use Committee (IACUC) of Sun Yat-sen University (SYSU-IACUC-2023-B0144). All mice were SPF-grade mice and were housed in an isolated space without pathogens. Animal group size was estimated based on power calculation (http://www.biomath.info/power). Mice were excluded from the study if they had no tumor, at the time of randomization. Tumor volumes were measured in a blinded fashion.

The mice used in the subcutaneous tumor model were 4-week-old male BALB/c nu/nu athymic mice purchased from GemPharmatech, Guangdong, China. H446 cells ($5 \times 10^6$) were suspended in 100 μL PBS/Matrigel (1:1) and subcutaneously injected into the mice. When the tumor volume reached approximately 70 mm$^3$, the animals were randomly divided into two groups ($n = 5$). The mice were then treated with vehicle or DN200434 (50 mg/kg, i.p.) seven times per week. The body weights of the mice and the volumes of the tumor were monitored every 3 days. The tumor volume was calculated using the equation V = π/6 (length × width$^2$). To evaluate the impact of ESRRG knockdown on tumor growth, H446 cells infected with lentivirus carrying either shRNA targeting ESRRG or control shRNA were injected subcutaneously into the dorsal flanks on both sides of the mice ($n = 9$ or 7). The process used to monitor tumor growth was similar in both treatment groups.

The mice used to establish a chemoresistant PDX tumour model were 4-week-old male BALB/c nu/nu mice purchased from

GemPharmatech, Guangdong, China. Human SCLC PDX LN140 was obtained from the West China Hospital of Sichuan University and subcutaneously transplanted into the BALB/c nu/nu mice. When the tumor volume was approximately 100 mm$^3$, the tumour-bearing mice were treated with multiple cycles of chemotherapy (Day 1, DDP 2.5 mg/kg, i.p.; Days 1–3, Eto 4 mg/kg, i.p.) until chemoresistance developed, as evidenced by no further reduction in tumor size with E/P treatment. We refer to this procedure as the EP protocol (Huang et al, 2020). When the original tumor-bearing mice exhibited weakness due to side effects of the chemicals with which they were treated, the tumors were dissected and passaged into new mice. The chemoresistant PDXs were propagated in the dorsal flanks of the mice on both sides. When the tumor volume reached approximately 50 mm$^3$, the mice were randomly divided into four groups. The groups were treated with vehicle, DN200434 (10 mg/kg, i.p.), Chemo (Day 1, DDP 2.5 mg/kg, i.p.; Days 1–3, Eto 4 mg/kg, i.p.) alone or in combination seven times per week ($n = 6$); 1 week was considered one cycle. Tumor volume and body weight were measured twice weekly. The volume was calculated using the equation V = π/6 (length × width$^2$). The mice were sacrificed at the end of the studies, and the tumors were harvested, weighed and subjected to further analysis.

For the study of liver metastasis in SCLC models (Fig. 3F), 4-week-old male NOD-SCID mice were purchased from GemPharmatech, Guangdong, China. A suspension of $2 \times 10^6$ H1048 cells in 25 μL of PBS was injected into the spleen of these animals ($n = 5$ per group). The wounds were closed using sterile medical sutures supplemented with 3 M Vetbond Tissue Adhesive. Following anaesthesia via isoflurane inhalation and intraperitoneal administration of D-luciferin, the mice were subjected to bioluminescence imaging using the IVIS Spectrum In Vivo Imaging System. Three days after inoculation, the mice were treated with DN200434 (25 mg/kg/day, i.p.). To assess tumor liver metastasis, weekly liver imaging of the mice was performed using the IVIS system. After 28 days, the mice were euthanized, and their livers were promptly removed, fixed and subjected to H&E staining.

For intravenous transplantation of SCLC models (Figs. 3H and 6F), 4-week-old male NOD-SCID mice were purchased from GemPharmatech, Guangdong, China. Suspensions of $2 \times 10^6$ H69AR cells in 100 μL of PBS were injected into the tail veins of the mice ($n = 6$ per group in Fig. 3 and $n = 5$ per group in Fig. 6). After induction of anaesthesia via isoflurane inhalation, in vivo bioluminescence imaging was performed using the IVIS Spectrum In Vivo Imaging System following intraperitoneal injection of D-luciferin. After inoculation with the cells, the mice were treated with vehicle, DN200434 (10 mg/kg/day, i.p.) and Chemo (Day 1, DDP 2.5 mg/kg, i.p.; Days 1–3, Eto 4 mg/kg, i.p.) alone or in combination; 1 week was considered one cycle. Tumor metastasis was evaluated by performing weekly imaging of the mice using the IVIS system. After 26 days, the mice were euthanized, and their organs were quickly removed, fixed and subjected to H&E staining.

## Bioinformatics analysis

Single-cell RNA sequencing data for SCLC patients were retrieved from the Human Tumor Atlas Network database (https://data.humantumoratlas.org. syn23593846). The sequencing data included samples from 21 human SCLC biospecimens. The Single-cell analysis was conducted following previously published protocols (Han et al, 2022). Subsequent analysis of the

**The paper explained**

**Problem**
SCLC is notorious for its aggressive behavior, early metastasis, and poor prognosis. Despite initial efficacy of conventional treatments such as chemotherapy and immunotherapy, SCLC frequently metastasises, leading to limited survival rates. Addressing metastasis and uncovering novel therapeutic targets is essential to improve prognosis and treatment outcomes for SCLC patients.

**Results**
Estrogen-related receptor gamma (ERRγ) was overexpressed in metastatic SCLC tumors, showing a positive correlation with disease progression. We uncovered a role for ERRγ in the regulation of extracellular matrix (ECM) remodeling and cell adhesion, pivotal processes in metastasis, through direct modulation of key genes implicated in these pathways. Suppression of ERRγ, both through genetic and pharmacological means, decreased collagen production, impaired cell-matrix adhesion, and suppressed microfilament formation, consequently inhibiting SCLC cell invasion and tumor metastasis. Furthermore, treatment with ERRγ antagonists suppressed tumor growth and metastasis, while concurrently reinstating chemosensitivity in resistant SCLC across diverse cell-derived and patient-derived xenograft models.

**Impact**
Our findings identify ERRγ as a pivotal player in ECM remodeling and a favourable therapeutic target in advanced SCLC. Given the druggability of ERRγ and the availability of potent ERRγ antagonists, these findings could expedite the development of new therapeutic modalities for advanced SCLC.

scRNA-seq data and GSEA was performed using R (version 3.4) with the edgeR package.

## Statistical analysis

All analyses were performed using GraphPad Prism 8.0 (GraphPad Software, USA). The data are presented as the mean ± s.d./s.e.m. of the values obtained in three independent experiments. Statistical analysis was conducted using two-tailed Student's $t$ tests to compare the means. $*P < 0.05$, $**P < 0.01$, $***P < 0.001$, and $****P < 0.0001$ were considered to indicate statistical significance. A $P$ value $> 0.05$ was considered not significant.

## Data availability

The gene expression data obtained in the RNA-seq analysis were deposited in the NCBI GEO database (GSE259271). ChIP-seq data were deposited in the NCBI GEO database (GSE259272).

The source data of this paper are collected in the following database record: biostudies:S-SCDT-10_1038-S44321-024-00108-z.

## Peer review information

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

## Acknowledgements

This research was supported by the National Natural Science Foundation of China (82273956, 82304520, 82373186), the Guangdong Basic and Applied Basic Research Foundation (2022B1515130008, 2022A1515010951), the Key Research and Development Plan of Guangzhou City (202206080007), the China Postdoctoral Science Foundation (2023M734065), the Fundamental Research Funds for the Central Universities, Sun Yat-sen University (No.23ptpy33) and the Science and Technology Planning Project of Guangdong Province (2023A0505010013).

## Author contributions

**Hong Wang**: Software; Funding acquisition; Validation; Project administration; Writing—review and editing. **Huizi Sun**: Data curation; Software; Validation; Writing—original draft; Writing—review and editing. **Jie Huang**: Writing—review and editing. **Zhenhua Zhang**: Data curation; Formal analysis; Validation; Visualization. **Guodi Cai**: Validation. **Chaofan Wang**: Conceptualization; Methodology. **Kai Xiao**: Conceptualization; Methodology. **Xiaofeng Xiong**: Conceptualization; Methodology. **Jian Zhang**: Conceptualization; Funding acquisition; Methodology. **Peiqing Liu**: Conceptualization; Methodology. **Xiaoyun Lu**: Supervision; Project administration. **Weineng Feng**: Supervision; Project administration. **Junjian Wang**: Supervision; Funding acquisition; Writing—original draft; Writing—review and editing.

Source data underlying figure panels in this paper may have individual authorship assigned. Where available, figure panel/source data authorship is listed in the following database record: biostudies:S-SCDT-10_1038-S44321-024-00108-z.

## Disclosure and competing interests statement

The authors declare no competing interests.

