## [Peer Review File · EMBO Molecular Medicine]

Therapeutic targeting ERR γ suppresses metastasis via extracellular matrix remodeling in SCLC

Junjian Wang, Hong Wang, Huizi Sun, Jie Huang, Zhenhu Zhang, Guodi Cai, Chaofan Wang, Kai Xiao, Xiaofeng Xiong, Jian Zhang, Peiqing Liu, Xiaoyun Lu, and Weineng Feng

Corresponding authors: Junjian Wang (wangjj87@mail.sysu.edu.cn) , Xiaoyun Lu (luxy2016@jnu.edu.cn), Weineng Feng (fwneng@fsyyy.com)

Review Timeline:

Submission Date:	26th Jan 24
Editorial Decision:	9th Feb 24
Revision Received:	20th May 24
Editorial Correspondence:	19th Jun 24
Authors' Correspondence:	20th Jun 24
Editorial Correspondence:	20th Jun 24
Editorial Decision:	25th Jun 24
Revision Received:	30th Jun 24
Accepted:	9th Jul 24

Editor: Lise Roth

Transaction Report:

9th Feb 2024

Dear Prof. Wang,

Thank you for the submission of your manuscript to EMBO Molecular Medicine. We have now received feedback from the three reviewers who agreed to evaluate your manuscript. As you will see from the reports below, the referees acknowledge the interest of the study and are overall supporting publication of your work pending appropriate revisions. In particular, the referees pointed out the lack of rigor in figures and methods presentation. As mentioned by referee #1, migration or transwell assays will need to be repeated within a shorter timeframe, and concerns on cell and metastasis models need to be addressed experimentally if possible, or by clearly stating the study limitations in the manuscript text.

Addressing the reviewers' concerns in full will be necessary for further considering the manuscript in our journal, and acceptance of the manuscript will entail a second round of review. EMBO Molecular Medicine encourages a single round of revision only and therefore, acceptance or rejection of the manuscript will depend on the completeness of your responses included in the next, final version of the manuscript. For this reason, and to save you from any frustrations in the end, I would strongly advise against returning an incomplete revision.

We are expecting your revised manuscript within three months, if you anticipate any delay, please contact us.

We require:

4) A .docx formatted letter INCLUDING the reviewers' reports and your detailed point-by-point responses to their comments. As part of the EMBO Press transparent editorial process, the point-by-point response is part of the Review Process File (RPF), which will be published alongside your paper.

5) A complete author checklist, which you can download from our author guidelines (<https://www.embopress.org/page/journal/17574684/authorguide#submissionofrevisions>). Please insert information in the checklist that is also reflected in the manuscript. The completed author checklist will also be part of the RPF.

6) Please note that all corresponding authors are required to supply an ORCID ID for their name upon submission of a revised manuscript.

7) It is mandatory to include a 'Data Availability' section after the Materials and Methods. Before submitting your revision, primary datasets produced in this study need to be deposited in an appropriate public database, and the accession numbers and database listed under 'Data Availability'. Please remember to provide a reviewer password if the datasets are not yet public (see <https://www.embopress.org/page/journal/17574684/authorguide#dataavailability>).

8) For data quantification: please specify the name of the statistical test used to generate error bars and P values, the number (n) of independent experiments (specify technical or biological replicates) underlying each data point and the test used to calculate p-values in each figure legend. The figure legends should contain a basic description of n, P and the test applied. Graphs must include a description of the bars and the error bars (s.d., s.e.m.). Please provide exact p values.

10) We replaced Supplementary Information with Expanded View (EV) Figures and Tables that are collapsible/expandable online. A maximum of 5 EV Figures can be typeset. EV Figures should be cited as "Figure EV1, Figure EV2" etc... in the text and their respective legends should be included in the main text after the legends of regular figures.

13) Author contributions: CRediT has replaced the traditional author contributions section because it offers a systematic machine readable author contributions format that allows for more effective research assessment. Please remove the Authors Contributions from the manuscript and use the free text boxes beneath each contributing author's name in our system to add specific details on the author's contribution. More information is available in our guide to authors.

16) As part of the EMBO Publications transparent editorial process initiative (see our Editorial at <http://embomolmed.embopress.org/content/2/9/329>), EMBO Molecular Medicine will publish online a Review Process File (RPF) to accompany accepted manuscripts.

In the event of acceptance, this file will be published in conjunction with your paper and will include the anonymous referee reports, your point-by-point response and all pertinent correspondence relating to the manuscript. Let us know whether you agree with the publication of the RPF and as here, if you want to remove or not any figures from it prior to publication. Please note that the Authors checklist will be published at the end of the RPF.

I look forward to receiving your revised manuscript.

Yours sincerely,

Lise Roth

***** Reviewer's comments *****

Referee #1 (Comments on Novelty/Model System for Author):

The systems used are not real metastasis models.

Referee #1 (Remarks for Author):

SCLC is a very aggressive pulmonary neuroendocrine malignancy associated with limited therapeutic advancement. Wang et al. find that ERR γ is highly expressed in metastatic SCLC, and can promote SCLC metastasis potentially through ECM remodeling. Genetic knockdown or pharmacological inhibition of ERR γ markedly reduces collagen production and inhibits SCLC metastasis. Moreover, ERR γ also highly expressed in chemotherapy resistant SCLC, and combination treatment of ERR γ inhibitor and chemotherapy displayed synergistic tumor growth inhibition. This work is interesting as it clarifies the function of ERR γ protein in SCLC malignant progression, and provides a novel therapeutic target for this refractory disease. However, the logic link, English writing as well as the understanding of the SCLC research field need to be further improved.

1. Although the finding is interesting, the manuscript suffers from the frequent lack of necessary description of material and technical details. For example, how many different human clinical samples were analyzed? From own collection or other public datasets? How was the H-Score in Figure H and Fig. S1C calculated? How was the organoid culture established for SCLC, similar to previous report (PMID: 33159857)? How was PDX model established? For LN140, the authors claimed it's chemotherapy resistant. How was this model established? From previous study (PMID: 35449308), the growth of chemotherapy resistant PDX is comparable the control group under EP treatment but it seems not the case for LN140 (Fig. 6H). Also, tissue number of the SCLC is inconsistent between the legend and figure (Fig. S1C).

2. Another issue is about the SCLC cell lines used in this study: all are adhesion cell lines. Previous report (PMID: 35967587) identified the NCAM^{hi}CD44^{low} subpopulation (the suspension cell in culture) but not as the NCAM^{low}CD44^{hi} (the adhesion type in culture) responsible for SCLC metastasis. Since the authors haven't used the spontaneous metastasis model, e.g., subcu. or orthotopic transplantation to detect distant organ metastasis, it remains interesting to at least check the expression of NCAM and CD44 in their system and discuss with the literatures.

3. For the wound healing assay, it's necessary to exclude the effect of ESRRG inhibition upon cell proliferation (Fig. 2A-B).

4. Other minor issues: please provide the toxicity of ERR γ inhibitors in this study, and the body weight of the mice with or without drug treatments. Please explain the difference about H69AR between Fig. 2E and 2F upon DN200434 treatment. Statistical analysis of Fig. 6I needs to be added. "They exhibited a positive correlation in chemo-resistance samples, indicating a potential role of ERR γ in the drug resistance of SCLC (Fig. 6A-C, Supplementary Fig. 6A-C)." There is no Supplementary Fig. 6A-C in the suppl materials. Inconsistency between figure and legends in Fig 2J, no ShESRRG#3!

Referee #2 (Comments on Novelty/Model System for Author):

Human samples, human cell lines, all good models
Rigor is a bit low with some missing controls and quantifications, and bar graphs are not displayed correctly

Referee #2 (Remarks for Author):

This manuscript by Wang, Sun, Huang, and colleagues, investigates the role of estrogen-related receptor gamma (ERR γ) in SCLC metastasis. Single-cell RNA sequencing of human tumors and metastases showed high levels of ERR γ compared to other nuclear receptors, as well as increased expression of ERR γ during metastasis. Loss- and gain-of-function of ERR γ in several human SCLC cell lines show that ERR γ is critical for tumor growth, in culture and in mice. Treatment with an ERR γ inhibitor (DN200434) also reduced metastatic burden in splenic or tail vein injections. RNA-seq and ChIP-seq/PCR analyses showed that loss of ERR γ leads to changes in gene programs associated with the ECM. Indeed, inhibition of ERR γ can lead to changes in collagen deposition and, overall, may create an environment facilitating metastasis. Finally, the authors also observed a correlation between ERR γ levels and chemoresistance and showed that ERR γ inhibition could enhance the effects of chemotherapy in chemoresistant models.

Overall, this is a well conducted study, which uses human primary samples and relevant cell line models to investigate ERR γ in SCLC. The data on migration/metastasis and chemoresistance are quite interesting and the phenotypes seem overall robust. This work should be of interest to a large group of investigators in the fields of SCLC and ERR γ . The main goal of the revisions should be to improve the clarity/rigor of some of the experiments/figures.

1) For all figures, the bar graphs should show the individual data points from independent experiments, not just the mean values. For each experiment, the number of mice should be clarified, and whether just one experiment was performed with several mice, or several independent experiments.

2) Figure 1D: difficult to see the "normal" (orange) cells

3) Figure 1H: do the authors have negative controls for the immunostaining (maybe from knock-down cells), to make sure that they are quantifying ERR γ ?

4) Can the authors correlate expression of ERR γ with expression of NFIB, which other groups have found to be important for metastasis? Is there a correlation between ERR γ expression and SCLC subtypes (ASCL1, NEUROD1, etc.)?

5) It may be useful in the knock-down and overexpression experiments to show the levels of ERR γ in the various cell lines used (at baseline), maybe from published RNA-seq data. It could also be useful to show the DepMap data for ERR γ knock-down or knockout in the SCLC cell lines that are analyzed on this web site.

6) Figure 2K, no body weight is shown. Figure 2L needs to be quantified. Is Figure S2L immunostaining for ERR γ or for another marker? If it is ERR γ , why are the levels decreasing, and can this be quantified?

7) S3F is not in the manuscript, and there are no Ki67 data associated with 3G. Same problem with S3G - this whole part needs to be clarified.

8) The authors rely on DN200434 for a large number of experiments and it may be important to show its specificity, at least in a couple of simple experiments: for instance, using ERR γ knock-down cells and testing if the inhibitor does not have additional effects? Or maybe the literature already shows clearly that this compound is highly specific (even at 10 μ M)?

9) Figure 6I should be quantified.

10) The chemotherapy treatment protocol should be clarified, as EP is very toxic in mice and daily injections seem to be a lot, but maybe the authors use doses lower than usual?

Referee #3 (Comments on Novelty/Model System for Author):

The detailed information on the experiment is not provided.
Ex gene expression analysis or in vivo model.

In cell migration and transwell assays, an experimental duration of 48-72 hours led to the proliferation of SCLC cells, which may not accurately reflect the real effect of ERR γ in cancer metastasis.

Referee #3 (Remarks for Author):

The authors in this manuscript showed that the ERR γ gene is highly overexpressed in tumors of small cell lung cancer (SCLC), particularly in metastatic sites. Inhibition of ERR γ through genetic or pharmacological means significantly reduces collagen production, cell-matrix adhesion, and microfilament production, and suppresses SCLC tumor cell growth and metastasis both in

vitro and in vivo. Furthermore, an ERR γ antagonist has been found to increase SCLC sensitivity to chemotherapy in multiple preclinical models. These findings may be an alternative approach to treating metastatic SCLC.

Major comments

1. The cell migration and transwell assays were inappropriate, an experimental duration of 48-72 hours led to the confounding of proliferation effect of SCLC cells, which may not accurately reflect the real effect of ERR γ in inhibiting cell migration/invasion and implication in suppressing cancer metastasis.
2. The authors reported that estrogen-related receptor gamma (ERR γ) is overexpressed in metastatic SCLC tumors and positively associated with SCLC progression. They also need to investigate ERR γ expression in NSCLC tumors or another cancer type, and after that, the ERR γ is the unique target in SCLC that may be confirmed.
3. Genetic and pharmacological inhibition of ERR γ can inhibit cell survival and metastasis. In tumor metastasis experiments, the treatment period is longer, even up to 2-4 weeks. How can we specifically demonstrate that ERR γ plays a role in metastasis in SCLC?
4. They may need to explain why they alter the cell type between tumor growth and metastasis model. (H446, H1084, H69AR, and H128).
5. The detailed information of Methods and Figure legends need to be described more completely. Ex. What is the frequency and dose of DN200434 or EP in Animal models? How many wells/samples for the statistical diagram? How many animals or wells (N) in the individual experiments?
6. The authors need to discuss what kind of ERR γ relative drug can be designed and discuss the advantages or disadvantages of the recent antagonist, DN200434 in the article.
7. Discuss why Orphan receptors are a good drugable target in introduction or discussion. What benefit is it?

Minor comments

1. The quality of the WB results is inadequate. The relative intensity of each protein also needs to be shown.
2. The knockdown efficiency of shRNA ERR γ also needs to be shown (Fig 2J). The sequences of siRNA and shRNA seem to be different.
3. The X-axis in sFig2J needs to be indicated.
4. The authors indicated the body weight of mice across each group was not significantly changed. But we do not see the result.
5. Although they showed the fluorescent ERR γ level in IFA (Fig5C and 5D), and DN200434 has been shown to suppress ERR γ expression, the authors must provide ERR γ levels in all IHC results.
6. Genetic and pharmacological inhibition of ERR γ to suppress COL6A1 seem to be different. In Fig 4F, the siERR γ almost decreases two forms of COL6A1, but in sFig4B in H128 cells, the DN200434 treatment only decreases the higher M.W. COL6A1. Why is there a discrepancy between these results?
7. In Fig 5A, 5B, 5E, and 5F may need to show the large field (more cells). And in Fig 5C, and 5D may need to show the small field (enlarged cell); after that, we can see the location of ERR γ in cancer cells.
8. We do not see the Supplementary Fig. 3F.

Response to reviewers

For your convenience, we highlight changes (mostly new data) in red in the revised manuscript including Supplementary data and Figure legends.

***** Reviewer's comments *****

Referee #1 (Comments on Novelty/Model System for Author):

The systems used are not real metastasis models.

Response: We thank the reviewer's comment. In this study, we utilized two metastasis models, namely tail vein injection and intrasplenic injection, to investigate the role of ERR γ in SCLC metastasis. The intrasplenic injection of cancer cells is a widely employed method to establish liver metastasis models for studying distant organ metastasis, while tail vein injection in mice is another well-established approach in metastasis research. Work to establish, by different approaches including orthotopic transplantation and genetically engineered mouse (GEM) models, the role of ERR γ in SCLC metastasis and progression is part of our near future studies. We sincerely hope that the respected reviewer agrees with our assessment and understands the direction of our ongoing research.

Referee #1 (Remarks for Author):

SCLC is a very aggressive pulmonary neuroendocrine malignancy associated with limited therapeutic advancement. Wang et al. find that ERR γ is highly expressed in metastatic SCLC, and can promote SCLC metastasis potentially through ECM remodeling. Genetic knockdown or pharmacological inhibition of ERR γ markedly reduces collagen production and inhibits SCLC metastasis. Moreover, ERR γ also highly expressed in chemotherapy resistant SCLC, and combination treatment of ERR γ inhibitor and chemotherapy displayed synergistic tumor growth inhibition. This work is interesting as it clarifies the function of ERR γ protein in SCLC malignant progression, and provides a novel therapeutic target for this refractory disease. However, the logic link, English writing as well as the understanding of the SCLC research field need to be further improved.

Response: We thank the reviewer for his/her positive remarks and valuable comments. We have carefully edited and improved the logic link, language as well as the understanding of the SCLC research field in the revised manuscript.

1. Although the finding is interesting, the manuscript suffers from the frequent lack of necessary description of material and technical details. For example, how many different

human clinical samples were analyzed? From own collection or other public datasets? How was the H-Score in Figure H and Fig. S1C calculated? How was the organoid culture established for SCLC, similar to previous report (PMID: 33159857)? How was PDX model established? For LN140, the authors claimed it's chemotherapy resistant. How was this model established? From previous study (PMID: 35449308), the growth of chemotherapy resistant PDX is comparable the control group under EP treatment but it seems not the case for LN140 (Fig. 6H). Also, tissue number of the SCLC is inconsistent between the legend and figure (Fig. S1C).

Response: We appreciate the valuable suggestions and apologize for the deficiency in material and technical details in our manuscript. We have thoroughly checked for such issues throughout the manuscript and added the necessary descriptions in the revised manuscript. Our responses to the specific comments are provided as follows.

First, we analyzed clinical samples from our own collection (Fig. 1H and Fig. S1C) as well as public datasets (Fig. 1A-1G and Fig. S1A-B), and have provided the exact clinical sample sizes in the Materials and Methods section or figure legends of the revised manuscript.

Second, for the H-Score in Figure 1H and Fig.S1C, we used Image J software to calculate the percentage of positive staining of IHC.

Third, the process of establishing PDOs is similar to the method reported in the previous report (PMID: 33159857), as described in the Materials and Methods section.

Forth, the method for establishing the SCLC PDX model and the corresponding chemo-resistant PDX model is similar to the one reported in the previous study (PMID: 35449308), as described in the Materials and Methods section.

For chemotherapy resistant PDX model LN140, we repeatedly treated LN140 PDX mouse model with E/P and eventually established the chemo-resistant PDX models (LN140R), as evidenced by no further significant reduction in tumor size with E/P treatment (see figure below). The detailed method for establishing the drug-resistant LN140 PDX has been supplemented in the Materials and Methods section, and the following is the graphical representation of our construction of drug-resistant LN140 (see figure below). Our results showed that while individual DN200434 and E/P treatments exhibited only modest inhibition of tumor growth in our LN140R models, the combined therapy of DN200434 and E/P exhibited a synergistic effect in inhibiting tumor growth. We would like to respectfully point out that that the statistical analysis suggests that EP alone did not significantly inhibit tumor growth,

indicating that LN140R tumors are resistant to the EP regimen, which aligns with previous literature findings (PMID: 35449308).

In addition, we apologized for the previous oversight in incorrectly issue number of the SCLC between the legend and figure, we corrected the problems in Fig. S1C.

2. Another issue is about the SCLC cell lines used in this study: all are adhesion cell lines. Previous report (PMID: 35967587) identified the NCAM^{hi}CD44^{low} subpopulation (the suspension cell in culture) but not as the NCAM^{low}CD44^{hi} (the adhesion type in culture) responsible for SCLC metastasis. Since the authors haven't used the spontaneous metastasis model, e.g., subcu. or orthotopic transplantation to detect distant organ metastasis, it remains interesting to at least check the expression of NCAM and CD44 in their system and discuss with the literatures.

Response: This is an interesting point. We detected the expression of NCAM and CD44 in SCLC cell lines used in this study and found that H1048 cells exhibited NCAM^{hi}CD44^{low} expression pattern. Since a recent significant study (PMID: 35967587) identified NCAM^{hi}CD44^{low} cells in transgenic mouse models (RP mouse model) as SCLC metastasizing cells, it is reasonable to infer that this cell line possesses strong metastatic potential. Furthermore, our choice of using this cell line for splenic injection to establish a model of liver metastasis is also supported by a previous report (PMID: 22989420) (Fig. 3F-3G). In contrast, we utilized the cell line H446, which exhibits high expression of NCAM and low expression of CD44, for subcutaneous tumor xenografts. As expected, we did not observe liver metastasis or metastasis to other distant organs in H446 xenograft models (Fig. 2J-2K and data not shown). However, it is intriguing that despite H69AR cells (derived from suspension H69 cell) being CD44-positive and NCAM-negative, we observed systemic metastasis after intravenous injection (Fig. 3H-3I and Fig. 6F-6G), which is consistent with previous report (PMID: 25456735). Since SCLC exhibits high heterogeneity, future investigations that assess the antagonist effect using GEM models, such as the RP mouse model, will be valuable in elucidating the role of ERR γ in SCLC metastasis. We have included a discussion on this point in the revised manuscript.

3. For the wound healing assay, it's necessary to exclude the effect of ESRRG inhibition upon cell proliferation (Fig. 2A-B).

Response: We agree with the reviewer's comment and performed more experiments to address the points. As shown in Figure 3A-D in the revised manuscript, we conducted a 24-hour treatment with inhibitors and lentivirus to eliminate the impact of ESRRG inhibition on cell proliferation. The results revealed a significant inhibition of cell invasion and migration upon ESRRG suppression.

4. Other minor issues: please provide the toxicity of ERRy inhibitors in this study, and the body weight of the mice with or without drug treatments. Please explain the difference

about H69AR between Fig. 2E and 2F upon DN200434 treatment. Statistical analysis of Fig. 6I needs to be added." They exhibited a positive correlation in chemo-resistance samples, indicating a potential role of ERR γ in the drug resistance of SCLC (Fig. 6A-C, Supplementary Fig. 6A-C)." There is no Supplementary Fig. 6A-C in the suppl materials. Inconsistency between figure and legends in Fig 2J, no ShESRRG#3!

Response: We appreciate the reviewer's valuable suggestion and provide our response as following.

Firstly, we provided the toxicity of ERR γ inhibitors in this study, as shown in Supplementary Fig. S2P and Fig. S5D in the revised manuscript, our new data showed that the ESRRG inhibitors exhibited no toxicity, and there was no significant difference in body weight between the control group and the treatment group in mice.

Secondly, we examined the effect of DN200434 treatment on H69AR by different approaches. Results shown in Figure 2E represents growth inhibition of DN200434 on H69AR cells, H69AR cells were treated with vehicle or DN200434 as indicated, after 96 hours, total viable cells were counted. Results shown in Figure 2F represents the survival inhibition of DN200434 on H69AR cells detected by colony formation assay, SCLC cells were treated with vehicle or the indicated concentrations of DN200434 for 14 days, after which colony formation was assessed. Due to the difference in the initial cell numbers and treatment durations between the two experiments, in Fig. 2E, the cell number per well in the 6-well plate was 150,000 cells with a drug treatment duration of 96 hours. In Fig. 2F, the cell number per well in the 6-well plate was also 5000 cells, but the drug treatment duration was 14 days. Therefore, the results indicate that although DN200434 exhibits significant inhibitory effects on H69AR cells in both experiments, it has a stronger inhibitory effect on clone formation.

Thirdly, we apologize for the missing statistical analysis on Fig. 6I, we now included the statistical analysis in the revised manuscript. Regarding the issue with Supplementary Fig. 6A-C should be Supplementary Fig. 5A-C, we now corrected the problems in the revised manuscript. In addition, we also corrected the identity annotation in Figure 2J in the revised manuscript.

Referee #2 (Comments on Novelty/Model System for Author):

Human samples, human cell lines, all good models

Rigor is a bit low with some missing controls and quantifications, and bar graphs are not displayed correctly

Response: We greatly appreciate the comments made by this reviewer. We find many of his/her comments being constructive in strengthening the manuscript. As reviewer's suggestion, we have made revisions to enhance the rigor of our work.

Referee #2 (Remarks for Author):

This manuscript by Wang, Sun, Huang, and colleagues, investigates the role of estrogen-related receptor gamma (ERR γ) in SCLC metastasis. Single-cell RNA sequencing of human tumors and metastases showed high levels of ERR γ compared to other nuclear receptors, as well as increased expression of ERR γ during metastasis. Loss- and gain-of-function of ERR γ in several human SCLC cell lines show that ERR γ is critical for tumor growth, in culture and in mice. Treatment with an ERR γ inhibitor (DN200434) also reduced metastatic burden in splenic or tail vein injections. RNA-seq and ChIP-seq/PCR analyses showed that loss of ERR γ leads to changes in gene programs associated with the ECM. Indeed, inhibition of ERR γ can lead to changes in collagen deposition and, overall, may create an environment facilitating metastasis. Finally, the authors also observed a correlation between ERR γ levels and chemoresistance and showed that ERR γ inhibition could enhance the effects of chemotherapy in chemoresistant models.

Overall, this is a well conducted study, which uses human primary samples and relevant cell line models to investigate ERR γ in SCLC. The data on migration/metastasis and chemoresistance are quite interesting and the phenotypes seem overall robust. This work should be of interest to a large group of investigators in the fields of SCLC and ERR γ . The main goal of the revisions should be to improve the clarity/rigor of some of the experiments/figures.

Overall response: We very much appreciate the comments made by this reviewer. We find many of his/her comments being constructive in strengthening the manuscript.

1) For all figures, the bar graphs should show the individual data points from independent experiments, not just the mean values. For each experiment, the number of mice should be clarified, and whether just one experiment was performed with several mice, or several independent experiments.

Response: We agree with the reviewer's suggestion. As shown in all revised figures, we have included the individual data points from independent experiments alongside the mean values in the bar graphs. We apologize for the missing information on the animal experiments. In the revised manuscript, the number of mice has been clarified in the Figure legends and Materials and Methods section. In addition, we performed one experiment using multiple mice, and we have included this information in Materials and Methods section of the revised manuscript.

2) Figure 1D: difficult to see the "normal" (orange) cells

Response: Thank you for pointing this out. Because our single-cell sequencing analysis revealed that ESRRG exhibits minimal expression in the normal tissues of these seven cell types, it becomes challenging for us to observe the “normal” (orange) cells in the image. Therefore, we labeled the data points with the corresponding tissue information (normal or SCLC tumor) from which each cell type was derived in the revised manuscript.

3) Figure 1H: do the authors have negative controls for the immunostaining (maybe from knock-down cells), to make sure that they are quantifying ERR γ ?

Response: We appreciate the valuable suggestion. We now included the immunostaining using cells with knockdown of ERR γ expression as negative controls to ensure that our Immunohistochemistry (IHC) accurately quantifies ERR γ . As shown in the Supplementary Figure 2L-2M, the Immunohistochemistry (IHC) data showed that the expression of ERR γ was significantly reduced in tumor tissues with knockdown of ERR γ .

4) Can the authors correlate expression of ERR γ with expression of NFIB, which other groups have found to be important for metastasis? Is there a correlation between ERR γ expression and SCLC subtypes (ASCL1, NEUROD1, etc.)?

Response: This is an interesting point. Regarding the correlation between the expression of ERR γ and NFIB in SCLC patient samples, we conducted an analysis using a recently published RNA-seq dataset from clinical SCLC tumors (Cell. 2024 Jan 4;187(1):184-203). Our analysis revealed no significant correlation between the expression of ERR γ and NFIB, as shown in the figure below. Additionally, we assessed the expression of NFIB in the cell lines used in our study based on a previous report (PMID: 27374332). In Figure A, we examined the protein expression of NFIB and ERR γ in all the cell lines and found no correlation between their expression levels. Furthermore, in our RNA-seq analysis, as depicted in Figure B, knocking down ERR γ did not significantly affect the mRNA levels of NFIB."

Furthermore, we conducted an analysis of the published RNA-seq dataset (Cell. 2024 Jan 4;187(1):184-203) to investigate the relationship between ERR γ expression and SCLC subtypes, namely ASCL1, NEUROD1, YAP1, and POU2F3. The results demonstrated no significant correlation between the expression levels of ERR γ and ASCL1, NEUROD1, YAP1, or POU2F3 (As shown in the below figure).

5) It may be useful in the knock-down and overexpression experiments to show the levels of ERR γ in the various cell lines used (at baseline), maybe from published RNA-seq data. It could also be useful to show the DepMap data for ERR γ knock-down or knockout in the SCLC cell lines that are analyzed on this web site.

Response: We appreciate the valuable suggestion. We conducted an analysis of DepMap data to investigate the expression levels of ERR γ in SCLC cell lines and the impact of ERR γ knock-down in these cell lines. The results revealed that ERR γ levels were moderate in the H446 and H1048 cells used in this study, immunoblotting readily detected ERR γ protein in all of the SCLC cell lines used. Additionally, analysis of DepMap data showed that ESRRG siRNA could effectively inhibit the viability of the majority of SCLC cells, including the H446 and H1048 cells. These results indicate that ERR γ is essential for the survival of SCLC cells. Considering that the DepMap analysis did not include all the cells we used and provided only limited

information, we do not feel comfortable to present the information in the manuscript. We sincerely hope that the reviewer agrees with our overall assessment.

A

B

6) Figure 2K, no body weight is shown. Figure 2L needs to be quantified. Is Figure S2L immunostaining for ERR γ or for another marker? If it is ERR γ , why are the levels decreasing, and can this be quantified?

Response: We thank the reviewer's kindly reminding. We have included a graph of mouse body weight (Figure S2P) in the revised supplementary materials. Additionally, we have quantified Figure 2L in the revised Figure. In Figure S2L, the immunostaining is for ERR γ , because *in vitro* experiments have shown that the compounds DN200434 can also reduce the protein expression of ERR γ in tumor cells (H446 and H128), we conducted subcutaneous tumor inhibition using H446 cells and administered the compound DN200434. Through immunohistochemistry (IHC) experiments, we verified that it can also reduce ERR γ protein expression *in vivo*. We have included the quantification in the revised Figure S2L.

7) S3F is not in the manuscript, and there are no Ki67 data associated with 3G. Same problem with S3G - this whole part needs to be clarified.

Response: Thank you for your kindly reminding. We apologize for the confusion caused by these erroneous labels. We have carefully reviewed the relevant content and made the necessary corrections in the revised manuscript.

8) The authors rely on DN200434 for a large number of experiments and it may be important to show its specificity, at least in a couple of simple experiments: for instance, using ERR γ knock-down cells and testing if the inhibitor does not have additional effects? Or maybe the literature already shows clearly that this compound is highly specific (even at 10 μ M)?

Response: We appreciate the valuable suggestion. To elucidate the specificity of DN200434 treatment, we used two different ESRRG siRNA to specifically silence ESRRG expression. The results demonstrated that the inhibitory effect of DN200434 on cell survival, even at a concentration of 10 μ M, was effectively blocked in cells treated with ESRRG siRNA compared to control cells. These findings suggest that DN200434 exerts its inhibitory effect on cell survival via specifically inhibiting ERR γ activity.

In addition, according to previous literature reports (PMID: 31010838), their *in vitro* binding assays also demonstrated that DN200434 exhibited promising ERR γ binding affinity ($IC_{50} = 0.040 \mu\text{mol/L}$) and comparable subtype selectivity with no prominent ERR α , ERR β affinity ($IC_{50} > 10, 1.330 \mu\text{mol/L}$, respectively). Furthermore, they validated DN200434 as a specific inhibitor of ESRRG through crystallographic studies of the DN200434-ESRRG complex.

9) Figure 6I should be quantified.

Response: We appreciate the valuable suggestion. We have added the quantification in revised Figure 6I.

10) The chemotherapy treatment protocol should be clarified, as EP is very toxic in mice and daily injections seem to be a lot, but maybe the authors use doses lower than usual?

Response: We apologized for our unclear description. We provided more detail information to clarify chemotherapy treatment protocol in the Materials and Methods section of the revised manuscript. Mice in the EP group were treated with cisplatin (DDP 2.5 mg/kg, intraperitoneally) at day 1 and etoposide (Eto 4 mg/kg, ip) at Days 1, 2, 3. One week was considered as one cycle. The treatment did not result in obvious body weight loss and visible toxicity in mice.

Referee #3 (Comments on Novelty/Model System for Author):

The detailed information on the experiment is not provided.
Ex gene expression analysis or *in vivo* model.

In cell migration and transwell assays, an experimental duration of 48-72 hours led to the proliferation of SCLC cells, which may not accurately reflect the real effect of ERR γ in cancer metastasis.

Response: We very much appreciate the comments made by this reviewer. We find many of his/her comments being constructive in strengthening the manuscript. We

provided more detailed information on the experiment and performed additional experiments to address the effect of ERR γ in cancer metastasis.

Referee #3 (Remarks for Author):

The authors in this manuscript showed that the ERR γ gene is highly overexpressed in tumors of small cell lung cancer (SCLC), particularly in metastatic sites. Inhibition of ERR γ through genetic or pharmacological means significantly reduces collagen production, cell-matrix adhesion, and microfilament production, and suppresses SCLC tumor cell growth and metastasis both in vitro and in vivo. Furthermore, an ERR γ antagonist has been found to increase SCLC sensitivity to chemotherapy in multiple preclinical models. These findings may be an alternative approach to treating metastatic SCLC.

Response: We thank the reviewer for his/her positive remarks and valuable comments.

Major comments

1. The cell migration and transwell assays were inappropriate, an experimental duration of 48-72 hours led to the confounding of proliferation effect of SCLC cells, which may not accurately reflect the real effect of ERR γ in inhibiting cell migration/invasion and implication in suppressing cancer metastasis.

Response: We appreciate the valuable suggestion. We performed additional experiments to repeat the assay at cells treated for 24 hours to eliminate the impact of ESRRG inhibition on cell proliferation. The results showed that both the ERR γ antagonist DN200434 and knockdown of ERR γ significantly inhibited cell migration and invasion following 24 hours of treatment.

2. The authors reported that estrogen-related receptor gamma (ERR γ) is overexpressed in metastatic SCLC tumors and positively associated with SCLC progression. They also need to investigate ERR γ expression in NSCLC tumors or another cancer type, and after that, the ERR γ is the unique target in SCLC that may be confirmed.

Response: This is an interesting point and important question to us too. We conducted an analysis of published datasets and observed that there were no significant changes in the expression of ESRRG (ERR γ) between NSCLC tumors and normal tissues. However, in contrast, the expression of ESRRG was found to be significantly higher in SCLC tumors compared to normal tissues (As shown in the below figure). Furthermore, in our patient tissue microarray, which included six cases of NSCLC (three adenocarcinomas and three squamous cell carcinomas), we observed that the expression of ERR γ in these NSCLC cases was similar to that in normal lung tissue, with no significant difference. Additionally, the expression of ERR γ in NSCLC was significantly lower compared to its expression in SCLC. These findings suggest that ERR γ may serve as a unique target in SCLC (As shown in the below figure). Given the limited information obtained, we do not feel comfortable to present the information in the manuscript. We sincerely hope that the reviewer agrees with our overall assessment.

3. Genetic and pharmacological inhibition of ERR γ can inhibit cell survival and metastasis. In tumor metastasis experiments, the treatment period is longer, even up to 2-4 weeks. How can we specifically demonstrate that ERR γ plays a role in metastasis in SCLC?

Response: This is an interesting point. To further investigate the role of ERR γ in metastasis in SCLC, we performed additional IHC analysis on metastasis marker genes in tumors obtained at the experimental endpoint of our tumor metastasis experiments. The results demonstrated that the ERR γ antagonist DN200434 significantly suppressed the expression of metastasis marker genes, including Vimentin and β -catenin, in tumors from both the liver metastasis model, where H1048 cells were injected into the spleen, and the lung metastasis model, where H69AR cells were injected via the tail vein. These data, combined with our in vitro results on migration and invasion, suggest that while we cannot exclude the potential influence of DN200434-induced suppression of cell survival in these experiments, they do indicate the significant role of ERR γ in SCLC metastasis.

4. They may need to explain why they alter the cell type between tumor growth and metastasis model.

Response: We thank the reviewer's comment. Given the heterogeneity of SCLC, we generated xenografts using SCLC cell lines with distinct features, including varying growth characteristics, invasive abilities, and tendencies for metastasis. In previous literature reports (PMID: 32554616), they used H446 cells for subcutaneous transplantation experiments, so we followed their subcutaneous tumor model and used H446 cells for subcutaneous transplantation. In another study (PMID: 23011677), they constructed a liver metastasis model by intravenous injection of H1048 cells, with a 100% incidence rate of liver metastasis. Therefore, we referred to their liver metastasis model and chose H1048 cells for the liver metastasis model. Additionally, another study (PMID: 24888228) reported using H69AR cells to construct a tumor metastasis model *via* intravenous injection, so we referenced their intravenous injection model. Thus, using multiple cell types can provide a more comprehensive understanding of the mechanisms underlying SCLC tumor development and

metastasis. We thus sincerely hope that the respectful reviewer shares our assessment on this.

5. The detailed information of Methods and Figure legends need to be described more completely. Ex. What is the frequency and dose of DN200434 or EP in Animal models? How many wells/samples for the statistical diagram? How many animals or wells (N) in the individual experiments?

Response: We sincerely apologize for the missing detailed information. In the revised manuscript, we have now included the frequency and dose of DN200434 and EP in our animal models. Moreover, we have presented each data point in the statistical graph along with the corresponding sample size. Additionally, we have clarified the number of mice used in the Materials and Methods section and figure legends of the revised manuscript. Furthermore, we have clearly labeled the “*n*” to indicate the sample size in each individual experiment.

6. The authors need to discuss what kind of ERR γ relative drug can be designed and discuss the advantages or disadvantages of the recent antagonist, DN200434 in the article.

Response: We appreciate the reviewer’s valuable suggestion. We have included a discussion on the design of ERR γ -related drugs, as well as a discussion on the advantages and disadvantages of the recent antagonist, DN200434, in the revised manuscript.

7. Discuss why Orphan receptors are a good druggable target in introduction or discussion. What benefit is it?

Response: We appreciate the valuable suggestion. We have included a discussion on this point in both the introduction and discussion sections of the revised manuscript.

Minor comments

1. The quality of the WB results is inadequate. The relative intensity of each protein also needs to be shown.

Response: We thank the reviewer for the suggestion. We conducted relative intensity analysis for each protein, and the quantitative results are indicated in the revised figure.

2. The knockdown efficiency of shRNA ERR γ also needs to be shown (Fig 2J). The sequences of siRNA and shRNA seem to be different.

Response: We very much agree with the reviewer’s comment and performed additional experiments to examine the knockdown efficiency of shRNA ERR γ . As

shown in Supplementary Figure 2N, the western blotting experiment on tumors showed that the protein expression of ERR γ was indeed knocked down. Additionally, our IHC experiment also indicated that ERR γ protein was knocked down *in vivo* (Figure S2L-M).

In addition, as pointed out by the reviewer, siRNA and shRNA sequences are different. Although both shRNA and siRNA are used in RNA interference techniques, they differ in design and mechanism of action. We employed different software to design siRNA and shRNA sequences, and the results demonstrated that both approaches effectively inhibited the expression of ERR γ .

3. The X-axis in sFig2J needs to be indicated.

Response: We appreciate your kindly reminding. We corrected it in the revised Supplementary Figure 2J.

4. The authors indicated the body weight of mice across each group was not significantly changed. But we do not see the result.

Response: We appreciate this reviewer to point out missing body weight information. we have included the body weight of mice in the revised figures (Figure S2P and Figure S5D).

5. Although they showed the fluorescent ERR γ level in IFA (Fig5C and 5D), and DN200434 has been shown to suppress ERR γ expression, the authors must provide ERR γ levels in all IHC results.

Response: We appreciated your valuable suggestion and performed additional IHC experiments to provide ERR γ levels in the revised manuscript (Figure S2L and Figure S5E).

6. Genetic and pharmacological inhibition of ERR γ to suppress COL6A1 seem to be different. In Fig 4F, the siERR γ almost decreases two forms of COL6A1, but in sFig4B in H128 cells, the DN200434 treatment only decreases the higher M.W. COL6A1. Why is there a discrepancy between these results?

Response: We appreciate the keen eye of this reviewer. We have also noticed this phenomenon and repeated WB assay with cell lysis collected from H128 cells

multiple times, obtaining similar results, i.e. only high dose DN200434 treatment decreases two forms of COL6A1 in H128 cells. This could be due to the specific cellular background or the differential selective regulation of inhibitors compared to siRNA. However, the exact reasons still require further investigation, which we think are beyond the scope of our current manuscript.

7. In Fig 5A, 5B, 5E, and 5F may need to show the large field (more cells). And in Fig 5C, and 5D may need to show the small field (enlarged cell); after that, we can see the location of ERR γ in cancer cells.

Response: Following the reviewer's suggestion, we have reorganized these panels in our revised manuscript.

8. We do not see the Supplementary Fig. 3F.

Response: Thank you for your kindly reminding. We apologize for the confusion caused by these erroneous labels. We have carefully reviewed the relevant content and made the necessary corrections in the revised manuscript.

19th June 2024

Dear Prof. Wang,

Thank you for your email and please accept my apologies for the delay in getting back to you as I was attending a conference.

As indicated in my previous email, the referees have provided their feedback and are overall positive on your revised manuscript. However, before I can proceed with acceptance, I do need to have all raw data, and for each experiment the number of technical and biological replicates. Could you please provide this information via email? Would you prefer to have the manuscript sent back to you?

With kind regards,

Lise Roth

Lise Roth, PhD

Senior Editor

EMBO Molecular Medicine

20th June 2024

Dear Dr. Roth,

Thank you very much for your response. In fact, we have already provided the 'raw data' when submitting the revised manuscript, which are located in the "Figure Source Data" files. We will check these raw data again, and make the necessary modifications to the manuscript according to your requirements (i.e. indicating the number of technical and biological replicates for each experiment and each figure panel). We will then send you the revised manuscript and the raw data file via email as soon as possible.

Thank you again for your help!

Sincerely,

Junjian Wang

20th June 2024

Dear Prof. Wang,

Thank you for your email, and thank you for providing the source data, which I have already checked. Our question relates to the number of technical vs. biological replicates, for instance for Figure 2: are the 3 columns in the SD 3 technical replicates, or 3 independent experiments (and if so, did you also perform technical replicates?).

A simple description of the N/n number for each figure panel would be sufficient at this point.

Thank you very much!

With kind regards,

Lise Roth

25th Jun 2024

Dear Prof. Wang,

Thank you for submitting your revised study, which was sent back to the three initial referees. As you will see below, they are overall satisfied with the revisions, and I will therefore be able to accept your manuscript once the following points will be addressed:

1/ Referees' comments: please address the remaining concerns from referee #2.

2/ Manuscript text:

- Please remove the red font, and only keep in track changes mode any new modification.
- Please provide up to 5 keywords.
- Methods:
 - o Patient samples: include a statement that the experiments conformed to the principles set out in the WMA Declaration of Helsinki and the Department of Health and Human Services Belmont Report.
 - o Cells: please indicate whether the cells were authenticated and tested for mycoplasma contamination.
 - o Mice: please indicate the housing and husbandry conditions.
 - o Statistics: please include a statement on blinding, size sample, randomization and inclusion/exclusion criteria.
 - o Data availability: Please remove "All associated data relevant to this study including any supplementary methods are available upon request from the corresponding authors." and "Expanded view data, supplementary information, appendices are available for this paper at." Please provide URLs for the deposited datasets.
- Author contributions: CRediT has replaced the traditional author contributions section because it offers a systematic machine-readable author contributions format that allows for more effective research assessment. Please remove the Authors Contributions from the manuscript and use the free text boxes beneath each contributing author's name in our system to add specific details on the author's contribution. More information is available in our guide to authors.
- Please rename "Completing financial interests" to "Disclosure statement and competing interests" (<https://www.embopress.org/competing-interests>).
- Please remove the sentence about peer review from the manuscript text.

3/ Figures:

- Please remove the red font from the Appendix file. Please note that you have the possibility to make some of your appendix figures Expanded View (EV) Figures and Tables that are collapsible/expandable online. A maximum of 5 EV Figures can be typeset. EV Figures should be cited as 'Figure EV1, Figure EV2' etc... in the text and their respective legends should be included in the main text after the legends of regular figures.
- For the figures that you do NOT wish to display as Expanded View figures, they should be bundled together with their legends in a single PDF file called *Appendix*, which should start with a short Table of Content. Appendix figures should be referred to in the main text as: "Appendix Figure S1, Appendix Figure S2" etc.
- Additional Tables/Datasets should be labeled and referred to as Table EV1, Dataset EV1, etc. Legends have to be provided in a separate tab in case of .xls files. Alternatively, the legend can be supplied as a separate text file (README) and zipped together with the Table/Dataset file.

- Please address the following queries from our data editors (if not already addressed):

1. Please note that the exact p values are not provided in the legends of figures 2a-c, e-f, j-k; 3b-d; 4e, h-i; 5a, f-g, i.
2. Although 'n' is provided, please describe the nature of entity for 'n' in the legends of figures 2a-c, e-g, l; 3a-e; 4e, h-i; 5a, c-d, f-i; 6e, g.
3. Please note that the scale bar needs to be defined for figures 5c-d.

4/ Thank you for providing Source Data for all figures. Please address the following:

- Figure 2: identical values were found in Fig. 2C, please double check. Fig. 2J/2K: please provide all values (for each mouse) (files attached). Please confirm that no technical replicates were performed (also for other in vitro experiments).
- Figure 3: identical values were found for Fig. 3A/3C, please double check.
- Immunofluorescence pictures: original individual channel pictures should be provided

5/ Checklist:

- please double check the subsection "Animal observed in or captured from the field", as I don't think it applies to your study.
- Please check that you do not need to fill in the "Core facilities" section.
- Please fill in the entire section "Experimental study design and statistics"
- Please fill in the subsection "Ethics/Studies involving human participants: ethics approval and Helsinki declaration"
- Please double check the subsection: "Data availability/computational models", as I'm not sure it applies to your study.

6/ Please note that all corresponding authors are required to supply an ORCID ID for their name upon submission of a revised manuscript. ORCID identifiers are currently missing for Prof. Weineng Feng and for Prof. Xiaoyun Lu.

7/ I introduced minor modifications to your Paper Explained, please let me know if you agree with the following or amend as you see fit:

Problem

SCLC is notorious for its aggressive behavior, early metastasis, and poor prognosis. Despite initial efficacy of conventional treatments such as chemotherapy and immunotherapy, SCLC frequently metastasises, leading to limited survival rates. Addressing metastasis and uncovering novel therapeutic targets is essential to improve prognosis and treatment outcomes for SCLC patients.

Results

Estrogen-related receptor gamma (ERR γ) was overexpressed in metastatic SCLC tumors, showing a positive correlation with disease progression. We uncovered a role for ERR γ in the regulation of extracellular matrix (ECM) remodeling and cell adhesion, pivotal processes in metastasis, through direct modulation of key genes implicated in these pathways. Suppression of ERR γ , both through genetic and pharmacological means, decreased collagen production, impaired cell-matrix adhesion, and suppressed microfilament formation, consequently inhibiting SCLC cell invasion and tumor metastasis. Furthermore, treatment with ERR γ antagonists suppressed tumor growth and metastasis, while concurrently reinstating chemosensitivity in resistant SCLC across diverse cell-derived and patient-derived xenograft models.

Impact

Our findings identify ERR γ as a pivotal player in ECM remodeling and a favourable therapeutic target in advanced SCLC. Given the druggability of ERR γ and the availability of potent ERR γ antagonists, these findings could expedite the development of new therapeutic modalities for advanced SCLC.

8/ I introduced minor changes to your synopsis text, please let me know if you agree with the following or amend as you see fit:

Metastasis remains the leading cause of mortality in patients with small cell lung cancer (SCLC). Through single-cell, bulk transcriptome analysis and gene functional studies, ERR γ was identified as a key player of extracellular matrix (ECM) remodelling and a promising target for metastatic SCLC.

- ERR γ was overexpressed in metastatic SCLC tumors and positively correlated with disease progression.
- ERR γ was identified as a key determinant of ECM-related gene expression in SCLC cells.
- Genetic and pharmacological suppression of ERR γ reduced collagen production, cell-matrix adhesion, and suppressed microfilament formation, thereby halting SCLC cell invasion and metastatic dissemination.
- ERR γ antagonists suppressed SCLC tumor growth and metastasis, and restored the sensitivity of resistant SCLC to chemotherapy.

Thank you for providing a nice synopsis picture, please upload it as a separate png/tiff/jpeg file 550 px wide x 300-600 px high, and make sure that the text remains legible.

9/ As part of the EMBO Publications transparent editorial process initiative (see our Editorial at <http://embomolmed.embopress.org/content/2/9/329>), EMBO Molecular Medicine will publish online a Review Process File (RPF) to accompany accepted manuscripts.

In the event of acceptance, this file will be published in conjunction with your paper and will include the anonymous referee reports, your point-by-point response and all pertinent correspondence relating to the manuscript. Let us know whether you agree with the publication of the RPF and as here, if you want to remove or not any figures from it prior to publication. Please note that the Authors checklist will be published at the end of the RPF.

I look forward to receiving your revised manuscript.

Yours sincerely,

Lise Roth

***** Reviewer's comments *****

Referee #1 (Remarks for Author):

The authors have answered my concerns and this manuscript is ready for the publication.

Referee #2 (Comments on Novelty/Model System for Author):

Statistical analyses need to be clarified (technical vs. biological replicates)

Referee #2 (Remarks for Author):

The authors did well in their responses to the Reviewers. I note three points that need to be clarified further.

- 1) Because of the acute effect of ERR γ knock-down or inhibition on cell survival and proliferation, it is very difficult to interpret the data in vivo as a role for ERR γ in metastasis (if cancer cells die or don't proliferate, they won't form metastases). The authors should be very clear on this point in the manuscript.
- 2) There is no mention of mycoplasma testing in the cell lines in the Methods. If such tests have not been done, then the authors should simply state it, but it is important to know as mycoplasma infections could affect the reproducibility of these observations in other labs.
- 3) The authors should still specify in the figure legends the number of independent experiments vs. the number of technical replicates (it is unclear when they write "n=3" if they mean technical triplicates or 3 independent experiments).

Referee #3 (Remarks for Author):

I have no further comment.

Response to Referees

***** Reviewer's comments *****

Referee #1 (Remarks for Author):

The authors have answered my concerns and this manuscript is ready for the publication.

Response: We very much appreciate the comments made by this Referee.

Referee #2 (Comments on Novelty/Model System for Author):

Statistical analyses need to be clarified (technical vs. biological replicates)

Response: We appreciated your valuable suggestion and clarified the statistical analyses (technical vs. biological replicates) in the revised figure legends.

Referee #2 (Remarks for Author):

The authors did well in their responses to the Reviewers. I note three points that need to be clarified further.

Response: We thank the reviewer for his/her positive remarks and valuable comments.

1) Because of the acute effect of ERR γ knock-down or inhibition on cell survival and proliferation, it is very difficult to interpret the data in vivo as a role for ERR γ in metastasis (if cancer cells die or don't proliferate, they won't form metastases). The authors should be very clear on this point in the manuscript.

Response: We thank the reviewer's comment. This is a general issue in the study of metastasis. To confirm the role of ERR γ in metastasis in SCLC, we performed additional IHC analysis on metastasis marker genes in tumors obtained at the experimental endpoint of our tumor metastasis experiments. The results demonstrated that the ERR γ antagonist DN200434 significantly suppressed the expression of metastasis marker genes, including MMP9 and β -catenin, in tumors from both the

liver metastasis model, where H1048 cells were injected into the spleen, and the lung metastasis model, where H69AR cells were injected via the tail vein. These data, combined with our *in vitro* results on migration and invasion, suggest that while we cannot exclude the potential influence of DN200434-induced suppression of cell survival in these experiments, they do indicate the significant role of ERR γ in SCLC metastasis. To clarify this point, we have included a statement in the Results section of the revised manuscript, specifically under the subheading "ERR γ inhibition suppresses SCLC cell invasion and tumor metastasis both *in vitro* and *in vivo*".

2) There is no mention of mycoplasma testing in the cell lines in the Methods. If such tests have not been done, then the authors should simply state it, but it is important to know as mycoplasma infections could affect the reproducibility of these observations in other labs.

Response: Thank you for your kindly reminding. In the Materials and Methods section of the revised manuscript, we indicated that the cells were authenticated and tested for mycoplasma contamination.

3) The authors should still specify in the figure legends the number of independent experiments vs. the number of technical replicates (it is unclear when they write "n=3" if they mean technical triplicates or 3 independent experiments).

Response: We appreciated your valuable suggestion and specified the number of independent experiments vs. the number of technical replicates in the revised figure legends.

Referee #3 (Remarks for Author):

I have no further comment.

Response: We very much appreciate the comments made by this Referee.

- Please note that an ORCID identifier is still missing for Weineng Feng (each individual has to do it for him/herself, and we can unfortunately not do it on authors' behalf).

Our response: Prof. Weineng Feng has updated his ORCID identifier. We now provide his updated ORCID identifiers (ORCID: 0009-0002-1521-5306). I have verified that the ID appears to be correct and have attached a screenshot for your reference. Please check and use this ORCID identifiers (ORCID: 0009-0002-1521-5306).

- Thank you for resizing the synopsis picture. However, we note that the text is very small now, could you please provide a version with bigger font?

Our response: Please find the synopsis picture with a bigger font in the attached files.

- Source data: thank you for checking and providing revised files. Please provide an explanation for these changes. Please also check the source data for figure 2A (H446 cells), and figure 2J/K, where duplications were also found. If changes are needed, please provide an explanation. Original individual channel pictures are missing for Figure 6D.

Our response: Thank you for your kindly reminding. We provide our response as follows.

First: Our explanation for the changes made to the provided revised files: We sincerely apologize for inadvertently placing the data incorrectly while preparing the source data. We have carefully examined our original data and made the necessary corrections.

Second: We have checked the source data for figure 2A (H446 cells) and figure 2J/K. They are correct, and no changes are needed.

Third: Please find original individual channel pictures of Figure 6D in the attached files.

9th Jul 2024

Dear Prof. Wang,

Thank you for submitting your revised files and providing additional Source Data. I am pleased to inform you that your manuscript is accepted for publication and is now being sent to our publisher to be included in the next available issue of EMBO Molecular Medicine!

Yours sincerely,

Lise Roth
